# Information-Tight Value-Loss Guarantees for Test-Time Committees in Cooperative MARL

## Abstract

Cooperative multi-agent reinforcement learning (MARL) deployments increasingly spend test-time compute through committees of policy checkpoints, seeds, or ensemble advisors that vote on each agent's action. We study how to certify the team value-loss of such a frozen agreement-gated committee controller relative to a fixed reference policy $\pi^{\text{ref}}$, from deployment-time information: statistics logged during execution — the occupancy $\mu$, the endorsement-failure probability $g$, and the fallback action identity — together with coordinate value-gaps that are exact in finite tabular models and conservatively bounded otherwise. This certifies a frozen controller; it is not policy learning. We show that per-agent marginal certification is invalid — its under-estimation grows linearly with team size and vanishes at $n = 1$ — and that a reference-prefix telescoping bound can strictly under-estimate the loss, so validity requires a joint occupancy-weighted certificate. Our main result is a range-aware information characterization: the return range gives an information-independent ceiling $R_{\max} = H\Delta_r$, and deployment information induces a chain $C_0 \geq C_1 \geq C_2$ over nested sets $\mathcal{I}_0 \preceq \mathcal{I}_1 \preceq \mathcal{I}_2$, yielding the guarantee $J(\pi^{\text{ref}}) - J(\pi_N^{\text{ctrl}}) \leq R_{\max} \wedge C_2 \leq R_{\max} \wedge C_1 \leq R_{\max} \wedge C_0$. In the clean regime $\eta = 0$ we prove profile-relative optimality over an explicit constructive witness class and pre-cap coordinate-local sharpness. The underlying carrier bound uses only $g$, so it is agnostic to the committee's dependence structure and covers arbitrarily correlated advisors; logging the executed fallback identity is what tightens $C_1$ to $C_2$. We then turn $C_2$ into a fresh-rollout, distribution-free $1 - \delta$ certificate with an explicit conservative value-bound construction and a matching rare-unit lower bound. Exact cooperative Markov games verify validity and tightness against dynamic-programming truth, a rollout-bridge experiment demonstrates valid certification under conservative value bounds, and an over-dispersion experiment shows that a binomial plug-in under-covers on correlated committees while the dependence-agnostic certificate stays valid. On a standard neural benchmark (MPE) without exact ground truth, the same rollout construction yields a certificate for a trained committee that is valid by construction and, under variance-adaptive concentration, informative.

## 1 Introduction

Recent test-time-compute work documents empirical gains from repeated sampling, voting, debate, and verifier-based aggregation over multiple candidates such as policy checkpoints, random seeds, or ensemble advisors (Wang et al., 2023; Du et al., 2024; Snell et al., 2025; Brown et al., 2024; Lifshitz et al., 2025). In the cooperative multi-agent setting studied here, modelled as a Markov game or a Dec-POMDP (Shapley, 1953; Littman, 1994; Oliehoek & Amato, 2016), we consider *coordinate-wise* committee execution: for each agent a committee of advisors votes on that agent's coordinate, and the joint action is assembled from the selected coordinates. This aggregation improves outputs empirically, but it does not tell a practitioner how much team value they might lose by deploying the committee controller instead of a trusted reference policy. This paper supplies that guarantee: a valid, deployment-time certificate of the committee controller's team value-loss relative to the reference.

**Problem.** Given deployment-time information, what is the tightest valid certificate of the team value-loss of a frozen cooperative-MARL committee controller? We answer this through a reference-relative certification problem. The controller is fixed, the reference policy $\pi^{\text{ref}}$ is fixed, and the goal is to certify the downside of the deployed controller, not to train or improve it.

**Why this is a MARL problem.** In cooperative MARL, independently reasonable coordinates may combine into a jointly bad action. This makes per-agent marginal certification invalid, a failure that has no single-agent analogue and can grow linearly with the number of agents.

**Why this is a sequential RL problem.** Even if each coordinate is certified relative to $\pi^{\text{ref}}$, the controller's own state and prefix occupancy need not match the reference occupancy. A bound that telescopes only along the reference prefix can under-estimate the true value loss. The correct certificate must be occupancy-weighted under the deployed controller and prefix-aware inside each joint action.

**What kind of guarantee this is.** Our certificate is reference-relative and is built on the finite-horizon performance-difference lemma (Kakade & Langford, 2002), specialized to the deployed controller's occupancy. It is therefore related to, but distinct from, three neighboring guarantee types. Safe and conservative policy improvement bound the value of a policy that is being *selected or improved* relative to a baseline (Schulman et al., 2015; Achiam et al., 2017; Laroche et al., 2019; Kumar et al., 2020); off-policy evaluation estimates the value of a target policy from logged data using importance weights (Thomas et al., 2015; Jiang & Li, 2016). In both, the object is a policy one is choosing or estimating. Here the controller is *frozen*: we neither train nor select it, and we ask only for a valid upper bound on its reference-relative downside from what deployment makes available. Section 2 details these boundaries.

**Information controls tightness.** We consider three nested information sets:

$$\mathcal{I}_0 = \sigma(\mu, g), \qquad \mathcal{I}_1 = \sigma(\mu, g, W), \qquad \mathcal{I}_2 = \sigma(\mu, g, W, W_{\text{fb}}),$$

where $\mu$ is the unit-occupancy law, $g$ is the endorsement-failure probability, $W$ is the worst coordinate swing, and $W_{\text{fb}}$ is the conditional swing of the actually executed fallback action. The corresponding information terms are

$$C_0 = nH\,\mathbb{E}_{U\sim\mu}[g(U)(H - t_U)\Delta_r], \qquad C_1 = nH\,\mathbb{E}_{U\sim\mu}[g(U)W(U)], \qquad C_2 = nH\,\mathbb{E}_{U\sim\mu}[g(U)W_{\text{fb}}(U)]. \quad (1)$$

Here the *unit* $u = (t, s, i, a_{<i})$ is a single coordinate-decision point, $\mu$ is its occupancy law (of total mass $nH$), and all of these symbols are defined formally in Section 3 and collected in Table 2. Because returns are bounded, the deployed certificate is the capped quantity $R_{\max} \wedge C_k$ with $R_{\max} = H\Delta_r$. The cap is information-independent; the deployment-time information enters through the pre-cap chain $C_0 \geq C_1 \geq C_2$. Of these quantities, the occupancy $\mu$ and the failure probability $g$ are logged during execution, as is the fallback action identity that separates $C_1$ from $C_2$; the swings $W$ and $W_{\text{fb}}$ are *not* directly observable but are coordinate value-gaps requiring evaluation of $Q^{\pi^{\text{ref}}}$, exactly in a tabular model and otherwise only up to a conservative confidence bound (Theorem 4). We are therefore precise throughout that the certificate combines deployment logs *with* value-evaluation quantities, rather than resting on logs alone.

**Why the characterization is tight.** The three nested certificates are not an arbitrary ladder: each gap is forced by a missing piece of information. Without occupancy weighting the certificate is not even valid (Proposition 2); without the fallback log it cannot fall below the worst-swing term $C_1$ (Corollary 3); and without a conservative value bound the numerical evaluation of $W, W_{\text{fb}}$ is only diagnostic rather than strictly valid (Theorem 4). The characterization identifies which additional information is responsible for each tightening.

**General carrier and the clean failure-side regime.** The results sit in three layers. Theorem 1 is a general carrier bound valid for any endorsement tolerance $\eta \geq 0$, carrying a success-side term (a non-reference endorsed action may still be slightly suboptimal) and a failure-side term. The information characterization then works in the clean regime $\eta = 0$: endorsement executes the reference coordinate, so the success-side

term vanishes and the hierarchy isolates the failure-induced downside — joint miscoordination and horizon compounding — matching controllers that defer to the reference action upon endorsement. For $\eta > 0$, Theorem 1 remains valid with the additional success-side term, which a stress check (Remark 2) shows is necessary; Propositions 3 and 4 give the corresponding range-capped certificate and its success-side information tightening. The pure failure-side characterization is stated for the clean regime and not for arbitrary soft-endorsement controllers.

**Contributions.** First, we prove that per-agent marginal certificates fail in cooperative MARL, with underestimation that grows linearly in team size (Proposition 1), and give a sequential prefix-drift counterexample showing that reference-prefix telescoping is not valid (Proposition 2). Second, we give a range-aware information characterization (Theorem 2): an unconditional validity chain, profile-relative optimality over an *explicit constructive witness class*, and pointwise pre-cap sharpness. The carrier is *agnostic to the committee's dependence structure*, and fallback-action logging is what tightens $\mathcal{I}_1$ to $\mathcal{I}_2$. Third, we turn the operational certificate into a finite-sample $\mathcal{I}_2$ certificate from fresh rollouts, with an *explicit conservative value-bound construction* that discharges the value-bound assumption and a matching rare-unit lower bound. Exact tabular experiments check validity against dynamic-programming truth, a rollout-bridge experiment demonstrates valid certification under conservative value bounds, an over-dispersion experiment confirms dependence agnosticism, and the same rollout construction yields an informative certificate for a trained committee on a neural benchmark (MPE).

## 2 Related Work

We position the paper by methodological boundary rather than chronology. Table 1 summarizes how the object of guarantee differs from neighboring literatures.

**Cooperative MARL and credit assignment.** Cooperative MARL is commonly formulated through Markov games (Shapley, 1953; Littman, 1994) or Dec-POMDPs (Oliehoek & Amato, 2016). Centralized-training/decentralized-execution methods learn value decompositions or counterfactual credit assignments, including VDN, QMIX, QTRAN, COMA, and variance-aware policy-gradient analyses (Sunehag et al., 2018; Rashid et al., 2018; Son et al., 2019; Foerster et al., 2018; Kuba et al., 2021), and are benchmarked on environments such as SMAC, MPE, and JaxMARL (Samvelyan et al., 2019; Lowe et al., 2017; Rutherford et al., 2024). These methods decompose value for learning; see Oroojlooy & Hajinezhad (2023) for a recent cooperative-MARL survey. We instead decompose a frozen controller's realized downside for certification. The closest technical lineage is the sequential or virtual-order decomposition used in multi-agent trust-region methods (Kuba et al., 2022; Wen et al., 2022; Zhong et al., 2024), but our use is deployment-time endorsement-failure accounting rather than training-time monotone improvement.

**Relation to the performance-difference lemma.** Our carrier (Theorem 1) is a finite-horizon performance-difference identity (Kakade & Langford, 2002) specialized to the deployed occupancy with a coordinate telescoping inside each joint action. The performance-difference lemma is thus the *vehicle*, not the contribution: the contribution is the information characterization built on top of it, identifying which deployment-time quantities suffice for which level of tightness, together with the dependence-agnostic and conservative-value constructions that make the resulting certificate operational.

**Safe RL, conservative RL, and off-policy evaluation.** Safe policy improvement and conservative RL provide guarantees for selecting or improving policies relative to a baseline (Kakade & Langford, 2002; Schulman et al., 2015; Achiam et al., 2017; Laroche et al., 2019; Kumar et al., 2020; Gu et al., 2024); off-policy evaluation estimates the value of a target policy from logged data (Thomas et al., 2015; Jiang & Li, 2016). These lines certify or estimate policies. They do not characterize which deployment-time quantities are sufficient for a reference-relative downside certificate of a fixed committee controller.

**Robustness certification in MARL.** Certified robustness work bounds behavioral or value degradation under observation or message perturbations (Mu et al., 2023; Yuan et al., 2024). This is adjacent in spirit

| Line of work | Object of guarantee | Contrast with this paper |
| --- | --- | --- |
| Safe / conservative policy improvement | policy selected or improved vs. baseline | certification of a fixed deployed controller |
| Conservative offline RL | value of a learned policy from logged data | no on-policy fresh-rollout certificate |
| Off-policy evaluation | value of a target policy via importance weights | no per-unit attribution; needs density ratios |
| MARL robustness certification | degradation under input/message perturbation | loss induced by committee gating |
| Trust-region MARL decomposition | training-time monotone improvement | deployment-time failure decomposition |
| Test-time compute / committees | empirical gains from voting/debate | no finite-sample MARL value-loss certificate |
| **This work** | **value-loss of a frozen committee controller** | **finite-sample certificate from deployment logs plus value evaluation** |

Table 1: Comparison of guarantee targets across neighboring literatures. The distinguishing feature is that we certify the reference-relative downside of a frozen cooperative-MARL committee controller from deployment-time information (logs and value-evaluated coordinate gaps).

but different in object: robustness certificates protect against input or message perturbations, whereas our certificate quantifies the team value-loss induced by agreement-gated committee execution itself.

**Test-time compute and committees.** Repeated sampling, voting, debate, and verifier-based aggregation improve outputs empirically in modern systems (Wang et al., 2023; Du et al., 2024; Snell et al., 2025; Brown et al., 2024; Lifshitz et al., 2025). These works motivate spending inference-time compute through committees, but they do not provide on-policy, finite-sample, reference-relative value-loss certificates for cooperative MARL controllers.

**Distribution-free finite-sample inference.** The finite-sample certificate uses bounded-variable concentration, especially empirical Bernstein bounds (Hoeffding, 1963; Maurer & Pontil, 2009; Boucheron et al., 2013), and the rare-unit lower bound uses a two-point Le Cam argument (Tsybakov, 2009). Distribution-free risk-control methods (Bates et al., 2021; Angelopoulos et al., 2024) are related in their finite-sample orientation, but their object is prediction-set risk rather than a frozen cooperative-MARL committee's team downside.

## 3 Problem Setup

Before the formal definitions we describe the objects informally and explain why each is needed; Figure 1 shows how they fit together and Table 2 collects the symbols. As stated in Definition 1 below, the state and action spaces and the horizon are finite, so the reference value function is well-defined and, in our exact experiments, computable by dynamic programming.

**Units and prefixes.** The committee acts one coordinate at a time: within a single joint action it fixes agent 1's coordinate, then agent 2's, and so on. To account for the loss incurred at each such step we index by a *unit* $u = (t, s, i, a_{<i})$: time $t$, state $s$, which agent $i$ is being decided, and the coordinates $a_{<i}$ already chosen for the earlier agents in this joint action. The prefix $a_{<i}$ matters because the committee's earlier coordinates may already have departed from the reference, so the value gap at agent $i$ must be measured *conditional on the prefix the controller actually produced*, not on the reference prefix. This prefix-awareness is exactly what a naive per-agent or reference-prefix bound misses, and Section 4 shows both misses can make such bounds invalid.

**Swings and the failure probability.** At a unit the committee either endorses (agrees on) a good coordinate or fails to. The *endorsement-failure probability* $g(u)$ is the probability of that failure. When failure occurs the controller falls back to some action; the resulting coordinate value-gap relative to the

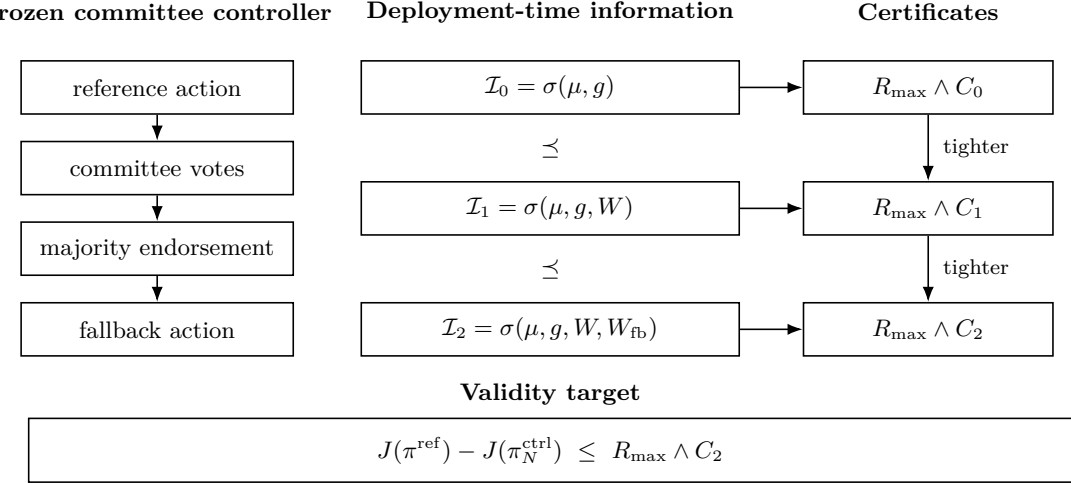

Figure 1: Certification architecture. A frozen agreement-gated committee controller (left) induces at each unit an occupancy law $\mu$ and an endorsement-failure probability $g$. Deployment-time information forms a refinement chain $\mathcal{I}_0 \preceq \mathcal{I}_1 \preceq \mathcal{I}_2$ (middle); each level yields a capped certificate $R_{\max} \wedge C_k$ (right), and richer information gives a tighter certificate. The range cap $R_{\max} = H\Delta_r$ is information-independent.

reference is the *fallback swing* $W_{\mathrm{fb}}(u)$, and the largest possible such gap over all actions is the *worst swing* $W(u) \geq W_{\mathrm{fb}}(u)$. Loss is thus accrued only on failure, weighted by $g(u)$, at a per-unit magnitude between $W_{\mathrm{fb}}(u)$ (what actually happens) and $W(u)$ (the worst that could happen).

**The information hierarchy.** How tight a certificate one can write depends on which of these quantities is available. With only the occupancy $\mu$ and the failure probability $g$, one must charge each failure the largest gap the remaining horizon allows, giving $C_0$. Knowing additionally the worst swing $W$ replaces that horizon-based bound by the true worst swing, giving the tighter $C_1$. Knowing additionally the realized fallback swing $W_{\mathrm{fb}}$ — which requires logging *which* fallback action was executed — replaces the worst swing by the swing that actually occurred, giving the tightest $C_2$. This is the chain $C_0 \geq C_1 \geq C_2$ of Equation (1). Because the true loss can never exceed the return range $R_{\max} = H\Delta_r$, each certificate is finally capped at $R_{\max}$.

**Definition 1** (Finite-horizon cooperative Markov game). An instance $M = (n, H, \mathcal{S}, \{\mathcal{A}_i\}, \{P_t\}, \{r_t\}, \rho_0)$ has $n$ agents, a finite horizon $H$, a finite state space $\mathcal{S}$, finite per-agent action spaces $\mathcal{A}_i$ with joint action space $\mathcal{A} = \prod_i \mathcal{A}_i$, time-dependent transitions $P_t(\cdot \mid s, a)$, shared reward $r_t(s, a) \in [0, \Delta_r]$, and initial distribution $\rho_0$. All theory in this paper is for such finite tabular models, in which $Q^{\pi^{\mathrm{ref}}}$ is well-defined and computable by dynamic programming; the learned-setting extension replaces exact values by conservative bounds (Theorem 4). Time-dependence may equivalently be encoded into the state. The team value is

$$J_M(\pi) = \mathbb{E}_\pi \left[ \sum_{t=0}^{H-1} r_t(s_t, a_t) \right].$$

We fix a reference policy $\pi^{\mathrm{ref}}$ with coordinate actions $a_i^{\mathrm{ref}}(s)$. A *unit* $u = (t, s, i, a_{<i})$ consists of time, state, coordinate index, and the executed action prefix. The coordinate $Q$-value $Q^{\pi^{\mathrm{ref}}}(u, a_i)$ is the value obtained by taking coordinate action $a_i$ at $u$ and then following $\pi^{\mathrm{ref}}$ for the remaining coordinates and future timesteps.

**Definition 2** (Reference-relative endorsed set and coordinate swing). For a unit $u$,

$$\Delta_+(u, a_i) = \left[ Q^{\pi^{\mathrm{ref}}}(u, a_i^{\mathrm{ref}}) - Q^{\pi^{\mathrm{ref}}}(u, a_i) \right]_+, \qquad G^\eta(u) = \{a_i : \Delta_+(u, a_i) \leq \eta\},$$

and $W(u) = \max_{a_i} \Delta_+(u, a_i)$. The positive part avoids assuming that $\pi^{\mathrm{ref}}$ is coordinate-greedy.

| Symbol | Meaning |
|---|---|
| $n$, $H$, $N$ | number of agents; (finite) horizon; committee size (odd) |
| $\mathcal{S}$, $\mathcal{A}_i$, $\mathcal{A}$ | finite state space; agent-$i$ action space; joint action space $\prod_i \mathcal{A}_i$ |
| $\Delta_r$, $R_{\max}$ | per-step reward range; return ceiling $R_{\max} = H\Delta_r$ |
| $\pi^{\mathrm{ref}}$, $\pi_N^{\mathrm{ctrl}}$ | reference policy; agreement-gated committee controller |
| $L$ | reference-relative team value-loss $J(\pi^{\mathrm{ref}}) - J(\pi_N^{\mathrm{ctrl}})$ |
| $u = (t, s, i, a_{<i})$ | unit: time, state, agent index, executed coordinate prefix |
| $Q^{\pi^{\mathrm{ref}}}(u, a_i)$ | coordinate $Q$-value: play $a_i$ at $u$, then follow $\pi^{\mathrm{ref}}$ |
| $\Delta_+(u, a_i)$ | clipped reference-relative disadvantage $[Q^{\pi^{\mathrm{ref}}}(u, a_i^{\mathrm{ref}}) - Q^{\pi^{\mathrm{ref}}}(u, a_i)]_+$ |
| $W(u)$, $W_{\mathrm{fb}}(u)$ | worst coordinate swing; conditional swing of the executed fallback |
| $g(u)$ | endorsement-failure probability $\mathbb{P}(F(u) = 1 \mid u)$ |
| $\alpha^\eta(u)$, $G^\eta(u)$ | endorsement probability; $\eta$-endorsed action set |
| $g_N(\alpha)$ | binomial majority-failure probability (i.i.d. special case) |
| $\psi(u)$, $w_\psi(u)$ | executed representative on endorsement; its swing ($= 0$ when $\eta = 0$) |
| $\mu$, $\bar{\mu}(u)$, $d_t^{\pi^{\mathrm{ctrl}}}$ | normalized unit law; unit-occupancy mass; controller state occupancy |
| $\mathcal{I}_0, \mathcal{I}_1, \mathcal{I}_2$ | nested information sets $\sigma(\mu, g) \preceq \sigma(\mu, g, W) \preceq \sigma(\mu, g, W, W_{\mathrm{fb}})$ |
| $C_0, C_1, C_2$ | information terms; deployed certificate is $R_{\max} \wedge C_k$ |
| $m$, $K$ | certification episodes; per-tail rollout budget |
| $\delta_B$, $\delta_G$ | sampling and value-bound failure probabilities |
| $B_Q$, $b_0$, $\widehat{B}_N$ | value-range bound; per-unit range $nHB_Q$; finite-sample certificate |
| $\omega(u)$ | generic $\mathcal{I}$-measurable per-unit weight in a certificate |
| $\mathfrak{M}_{\mathrm{tail}}$, $\mathfrak{M}_{\mathrm{R}}$ | constructive tail witness class; rich law-level class (Definition 9) |
| $\Theta_u$ | latent endorsement level of an exchangeable committee (Corollary 2) |
| $m_F$, $\delta'$, $\mathrm{rad}(K, \delta')$ | logged failed units; per-unit confidence level; rollout radius (Theorem 4) |

Table 2: Notation used throughout the paper.

**Definition 3** (Agreement-gated committee controller). At each unit $u$ the controller runs an *endorsement test* with failure event $F(u)$, whose probability we write as

$$g(u) = \mathbb{P}\big(F(u) = 1 \mid u\big) \in [0, 1].$$

On endorsement ($F = 0$) the controller executes a representative endorsed coordinate $\psi(u) \in G^\eta(u)$, incurring swing $w_\psi(u) = \Delta_+(u, \psi(u)) \leq \eta$. On failure ($F = 1$) it executes a fallback action $a^{\mathrm{fb}}(u)$, whose conditional swing is

$$W_{\mathrm{fb}}(u) = \mathbb{E}\big[\Delta_+(u, a^{\mathrm{fb}}) \mid u, F = 1\big], \qquad 0 \leq W_{\mathrm{fb}}(u) \leq W(u).$$

The controller is thus specified by the failure probability $g$, the endorsement swing $w_\psi$, and the fallback swing $W_{\mathrm{fb}}$, and every carrier and validity result below uses only these. No independence among advisors is assumed at this level. In the clean regime $\eta = 0$, every endorsed representative satisfies $\Delta_+(u, \psi(u)) = 0$, hence $w_\psi = 0$ and the downside is carried entirely by $g$ and $W_{\mathrm{fb}}$.

**Definition 4** (Conditionally-i.i.d. (binomial) committee: a special parameterization). The most common committee instantiates the failure probability of Definition 3 by a majority vote. If $N$ odd advisors propose coordinates that are conditionally i.i.d. given $u$, each endorsing with probability $\alpha^\eta(u) = \mathbb{P}\{a_i \in G^\eta(u) \mid u\}$, then majority failure has the binomial form

$$g_N(\alpha) = \mathbb{P}\{\mathrm{Bin}(N, \alpha) \leq \lfloor N/2 \rfloor\}, \qquad g(u) = g_N(\alpha^\eta(u)).$$

This parameterization is used only where noted — the budget-monotonicity Corollary 4 and the anchored threshold Lemma 1; every other result depends on $g$ alone and therefore holds for arbitrarily correlated or non-identically-distributed advisors, as made explicit in Corollary 2.

**Definition 5** (Unit occupancy). The controller induces a unit-occupancy mass

$$\bar{\mu}(u) = d_t^{\pi^{\mathrm{ctrl}}}(s)\mathbb{P}(a_{<i} \mid t, s),$$

whose total mass over all units is $nH$. Let $\mu = \bar{\mu}/(nH)$ be the normalized unit law. We write certificates in the form

$$C = nH\, \mathbb{E}_{U\sim\mu}[g(U)\omega(U)] = \sum_u \bar{\mu}(u)g(u)\omega(u).$$

For every admissible unit law, $0 \le \bar{\mu}(u) \le 1$.

**Definition 6** (Deployment-time information sets)**.** The deployment-time information sets are

$$\mathcal{I}_0 = \sigma(\mu, g) \preceq \mathcal{I}_1 = \sigma(\mu, g, W) \preceq \mathcal{I}_2 = \sigma(\mu, g, W, W_{\mathrm{fb}}).$$

*Remark* 1 (What is logged versus what is value-evaluated)*.* The three sets split into two kinds of quantity, which qualifies the word "observable":

$$\text{deployment logs: } (\mu,\ g,\ \text{fallback identity}), \qquad \text{value-evaluated quantities: } (W,\ W_{\mathrm{fb}}).$$

The logs are recorded during execution at negligible overhead. The swings $W$ and $W_{\mathrm{fb}}$ are coordinate value-gaps and require evaluating $Q^{\pi^{\mathrm{ref}}}$: exactly in a finite tabular model, and otherwise only up to a conservative confidence bound (Theorem 4). Strictly speaking, then, $\mathcal{I}_1$ and $\mathcal{I}_2$ are not log-only information sets: they combine deployment logs with value-evaluated coordinate gaps, and the strict-validity guarantees hold under exact evaluation or under the conservative bound — a point we return to in the finite-sample treatment (Section 7) and the limitations (Section 10).

**Definition 7** (Profile-relative optimal valid certificate)**.** For an information set $\mathcal{I}$, a $\mathcal{I}$-measurable certificate $\bar{C}$ is valid over a model class $\mathfrak{M}$ if $\bar{C} \ge J_{M'}(\pi^{\mathrm{ref}}) - J_{M'}(\pi^{\mathrm{ctrl}})$ for every $M' \in \mathfrak{M}$ consistent with the observed profile. The optimal certificate is

$$\mathcal{C}^{\mathrm{opt}}_{\mathfrak{M}}(\mathcal{I}) = \sup_{M'\in\mathfrak{M}:\ \mathrm{prof}_{\mathcal{I}}(M')=\mathrm{prof}_{\mathcal{I}}(M)} \left[ J_{M'}(\pi^{\mathrm{ref}}) - J_{M'}(\pi^{\mathrm{ctrl}}) \right].$$

The following threshold quantifies how enlarging the committee budget lowers the failure probability; its unanchored form drives the budget-monotonicity result of Section 8.

**Lemma 1** (Anchored eligibility threshold)**.** *Under the anchored protocol, where the reference action is a seated self-endorsing member and the remaining $N-1$ advisors endorse independently with probability $\alpha$, the anchored failure probability*

$$h_N(\alpha) = \mathbb{P}\left\{ \mathrm{Bin}(N-1,\alpha) \le \frac{N-3}{2} \right\}$$

*is strictly decreasing along odd $N \mapsto N+2$ if and only if $\alpha > (N+1)/(2N)$, with stationarity at $\alpha^\star_N = (N+1)/(2N)$. The unanchored threshold is the limit $1/2$. The proof is in Appendix A.7.*

## 4  Failure of Marginal and Reference-Prefix Certificates

**Proposition 1** (Marginal certificate failure and linear amplification)**.** *There exist a cooperative game and an agreement-gated controller for which the per-agent marginal certificate is strictly below the true team value-loss. On a canonical family, the ratio of true loss to marginal certificate is $(1-g) + ng$, hence grows linearly in $n$ and equals $1$ at $n = 1$.*

*Proof.* The full computation is in Appendix A.2. For $n = 2$, $H = 1$, actions $\{0, 1\}$, reference $(0, 0)$, fallback 1, and rewards $R(0,0) = 1$, $R(1,0) = R(0,1) = 0.9$, $R(1,1) = 0$, independent failures with probability $g$ give true loss $0.2g(1-g) + g^2$ and marginal certificate $0.2g$. At $g = 0.317$, the true loss is larger, so the marginal certificate is invalid. The $n$-agent quadratic-loss family gives true loss $g(1-g)/n + g^2$ and marginal certificate $g/n$, yielding ratio $(1-g) + ng$; the closed form matches this law exactly (Appendix B, Figure 12). $\qquad\square$

**Proposition 2** (Sequential prefix-drift under-estimation)**.** *There exist a time-inhomogeneous cooperative Markov game with $H \ge 2$ and a controller for which a reference-prefix telescoping bound strictly under-estimates the true loss, while the joint occupancy-weighted certificate $C_2$ remains valid.*

*Proof.* The construction and exact numbers are in Appendix A.2. Action-dependent transitions place positive controller occupancy on non-reference prefixes, where coordinate swings differ from the reference-prefix swings. Exact dynamic programming on the constructed instance gives true loss 0.21290, reference-prefix bound 0.20947, and joint certificate $C_2 = 0.21489$. The strict under-estimate is deterministic, not a sampling artifact. □

## 5 Occupancy-Weighted Value-Loss Carrier

The next theorem is the carrier inequality behind all certificates. It is a finite-horizon performance-difference decomposition under the deployed controller's occupancy, with a coordinate telescoping inside each joint action. It is related to approximate dynamic-programming error-propagation analyses (Munos & Szepesvári, 2008; Farahmand et al., 2010; Scherrer et al., 2015), but the decomposition here is over agent coordinates and endorsement failures.

**Theorem 1** (General carrier bound). *For any agreement-gated committee controller,*

$$J(\pi^{\mathrm{ref}}) - J(\pi_N^{\mathrm{ctrl}}) \leq nH \, \mathbb{E}_{U \sim \mu}\big[(1 - g(U))w_\psi(U) + g(U)W_{\mathrm{fb}}(U)\big].$$

*Proof.* See Appendix A.3 for the formal proof. The proof applies the finite-horizon performance-difference identity under $d_t^{\pi^{\mathrm{ctrl}}}$, expands the joint-action difference by replacing coordinates along the executed prefix distribution, and upper-bounds every coordinate increment by its clipped reference-relative disadvantage. Success contributes at most $w_\psi$, and failure contributes conditional mean $W_{\mathrm{fb}}$. □

Before turning to the clean regime, we package the general carrier into an explicit, capped certificate for arbitrary $\eta$, and show how logging the endorsed representative's identity sharpens it — mirroring the failure-side refinement of Section 6.

**Proposition 3** (Range-capped certificate for general $\eta$). *For any endorsement tolerance $\eta \geq 0$ and any agreement-gated committee controller,*

$$J(\pi^{\mathrm{ref}}) - J(\pi_N^{\mathrm{ctrl}}) \ \leq \ R_{\max} \wedge \big(S_\eta + C_2\big),$$

*where $S_\eta = nH \, \mathbb{E}_{U \sim \mu}[(1 - g(U))\,\eta]$ and $C_2 = nH \, \mathbb{E}_{U \sim \mu}[g(U)\,W_{\mathrm{fb}}(U)]$. As $\eta \to 0$ the success-side term $S_\eta$ vanishes and the bound recovers the clean-regime certificate of Corollary 1.*

*Proof.* From the carrier bound (Theorem 1), $L \leq nH \, \mathbb{E}[(1 - g)w_\psi + gW_{\mathrm{fb}}]$. Since $\psi(u) \in G^\eta(u)$ gives $w_\psi(u) \leq \eta$ pointwise, $\mathbb{E}[(1 - g)w_\psi] \leq \eta \, \mathbb{E}[1 - g]$, so $L \leq S_\eta + C_2$; capping by $R_{\max} \geq L$ preserves validity. □

**Proposition 4** (Success-side information tightening). *Let $S_\psi = nH \, \mathbb{E}_{U \sim \mu}[(1 - g(U))\,w_\psi(U)]$. For every $\eta \geq 0$,*

$$L \ \leq \ R_{\max} \wedge \big(S_\psi + C_2\big) \ \leq \ R_{\max} \wedge \big(S_\eta + C_2\big),$$

*and the first inequality is tight in the same sense as the failure-side chain: logging the identity of the executed endorsed representative resolves its realized swing $w_\psi$ and tightens the coarse tolerance term $S_\eta$ to $S_\psi$ (witness in Appendix A.4).*

*Proof.* The refinement is the carrier bound capped by $R_{\max}$; $S_\psi \leq S_\eta$ is pointwise from $w_\psi \leq \eta$. The tightness witness is deferred to Appendix A.4. □

**Corollary 1** (Clean agreement-gated baseline). *When $\eta = 0$ and majority endorsement executes the reference coordinate, $w_\psi = 0$ and*

$$J(\pi^{\mathrm{ref}}) - J(\pi_N^{\mathrm{ctrl}}) \leq nH \, \mathbb{E}_{U \sim \mu}[g(U)W_{\mathrm{fb}}(U)] \leq nH \, \mathbb{E}_{U \sim \mu}[g(U)W(U)].$$

**Corollary 2** (Agnosticism to committee dependence structure). *The carrier bound and the validity chain (Theorem 2(a)) use only the deployment-time failure probability $g(u) = \mathbb{P}(F(u) = 1 \mid u)$ and the conditional swing $W_{\mathrm{fb}}$; they do not use the binomial form $g_N(\alpha)$. Consequently they hold for any committee whose advisors are arbitrarily correlated or non-identically distributed, once $g$ is defined or estimated. In particular, if the $N$ endorsement votes are exchangeable Bernoulli with a latent endorsement level $\Theta_u$, so that conditionally $Y_1, \ldots, Y_N \mid \Theta_u$ are i.i.d. Bernoulli($\Theta_u$), then the majority-failure probability is the mixture*

$$g(u) = \mathbb{E}_{\Theta_u}\big[\mathbb{P}\{\mathrm{Bin}(N, \Theta_u) \leq \lfloor N/2 \rfloor \mid \Theta_u\}\big],$$

*and the carrier holds verbatim with this $g$. The binomial i.i.d. form is the degenerate case $\Theta_u \equiv \alpha^\eta(u)$; it is needed only for the budget-monotonicity Corollary 4 and the anchored threshold Lemma 1.*

*Proof.* Inspection of the proof of Theorem 1: the only property used at unit $u$ is that the executed action is the reference coordinate with probability $1 - g(u)$ (incurring at most $w_\psi$) and a fallback with probability $g(u)$ (incurring conditional mean $W_{\mathrm{fb}}$). The binomial form enters only when translating an endorsement rate $\alpha$ into $g$. The exchangeable case is the de Finetti representation of an exchangeable binary vote sequence; majority failure is the stated mixture. $\qquad\square$

*Remark 2* (The success term is not removable). For $\eta > 0$, the pure failure term need not upper-bound the total downside because a successful non-reference representative may still incur $w_\psi > 0$. In the reported stress check at $\eta = 0.2$, dropping the success-side term makes the pure failure term fall below the true loss in every instance (Figure 10).

## 6 Range-Aware Information Characterization

Let $L = J(\pi^{\mathrm{ref}}) - J(\pi_N^{\mathrm{ctrl}})$ and $R_{\max} = H\Delta_r$. Recall from Equation (1) the information terms

$$C_0 = nH\,\mathbb{E}_\mu[g(U)(H - t_U)\Delta_r], \qquad C_1 = nH\,\mathbb{E}_\mu[g(U)W(U)], \qquad C_2 = nH\,\mathbb{E}_\mu[g(U)W_{\mathrm{fb}}(U)].$$

The bounded return range gives $L \leq R_{\max}$ independent of the available information.

**Definition 8** (Pre-cap coordinate-local certificate class). For an information set $\mathcal{I}$, define

$$\mathcal{C}_{\mathrm{coord,pre}}(\mathcal{I}) = \left\{\sum_u \bar{\mu}(u)g(u)\omega(u) : \omega \text{ is } \mathcal{I}\text{-measurable and coordinate-local}\right\}.$$

This class compares the local information terms before applying the final global cap; *coordinate-local* means $\omega(u)$ depends on $u$ only through the $\mathcal{I}$-measurable per-unit quantities at $u$, with no coupling across units.

**Definition 9** (Constructive tail witness class). Let $\mathfrak{M}_{\mathrm{tail}}$ be the class of finite-horizon cooperative Markov games with rewards in $[0, \Delta_r]$ and agreement-gated controllers generated by two explicit gadgets: (i) the *finite-tail gadget* of Lemma 4, which realizes a prescribed per-unit coordinate gap $\omega(u) \in [0, (H - t)\Delta_r]$ over the remaining horizon with additive, non-overlapping losses; and (ii) the *cliff-chain gadget* of Lemma 5, which saturates the realized loss at $R_{\max}$ while inflating the worst-case and range certificates above $R_{\max}$. The class $\mathfrak{M}_{\mathrm{tail}}$ is closed under disjoint composition of these gadgets across units.

**Assumption 1** (Rich law-level witness class). $\mathfrak{M}_{\mathrm{tail}} \subseteq \mathfrak{M}_{\mathrm{R}}$, where $\mathfrak{M}_{\mathrm{R}}$ is any model class that reproduces, for each admissible law-level profile $(\mu, g, \omega)$ with $0 \leq g(u) \leq 1$, $0 \leq \omega(u) \leq (H - t_u)\Delta_r$ and a self-consistent unit law $\mu$, the profile quantities required by the corresponding information set, and is closed under the two gadgets of Definition 9. This abstraction is used only to phrase the lower bound at the level of a model class; the constructive content is carried entirely by $\mathfrak{M}_{\mathrm{tail}}$.

The next theorem separates the population-level characterization into three logically distinct parts. Part (a) is the deployable validity guarantee and requires only bounded rewards. Part (b) is a profile-relative optimality statement over the constructive witness class. Part (c) isolates pre-cap coordinate-local sharpness. Finite-sample estimation of the operational certificate is handled separately in Section 7.

**Theorem 2** (Range-aware information characterization). *In the clean agreement-gated regime $w_\psi = 0$:*

(a) **Validity chain, unconditional.**

$$L \leq R_{\max} \wedge C_2 \leq R_{\max} \wedge C_1 \leq R_{\max} \wedge C_0.$$

(b) **Constructive profile-relative optimality.** *Over the explicit class $\mathfrak{M}_{\text{tail}}$,*

$$\mathcal{C}^{\text{opt}}_{\mathfrak{M}_{\text{tail}}}(\mathcal{I}_k) = R_{\max} \wedge C_k, \qquad k = 0, 1, 2,$$

*hence the same equality holds over any $\mathfrak{M}_{\text{R}} \supseteq \mathfrak{M}_{\text{tail}}$ satisfying Assumption 1. The cap-inactive case $(C_k \leq R_{\max})$ is realized by the finite-tail gadget for all $k$; the cap-active case $(C_k > R_{\max})$ is realized explicitly by the cliff-chain gadget for $k \in \{0, 1\}$. For $k = 2$ the optimality is reached entirely through the cap-inactive branch: every admissible instance in our construction satisfies $C_2 \leq R_{\max}$ (Remark 4), so the theorem does not rely on a separate cap-active $\mathcal{I}_2$ witness.*

(c) **Pre-cap coordinate-local sharpness.** *For every admissible unit law,*

$$\mathcal{C}^{\text{opt}}_{\text{coord,pre}}(\mathcal{I}_k) = C_k, \qquad k = 0, 1, 2.$$

*No coordinate-local pre-cap certificate can reduce the pointwise weight used by $C_k$ on a positive-mass set while remaining valid.*

*Proof.* Part (a) follows from Corollary 1, $W_{\text{fb}} \leq W \leq (H - t)\Delta_r$, and $L \leq R_{\max}$. For part (b), part (a) gives the upper bound; the lower bound is supplied by the constructive gadgets (Lemmas 4, 5) in Appendix A.5. For part (c), admissibility gives $\bar{\mu}(u) \leq 1$, $g(u) \leq 1$, and $\omega_k(u) \leq (H - t_u)\Delta_r \leq R_{\max}$, so every single-unit witness is cap-inactive at the unit level. A certificate that undercuts the local weight on any positive-mass set is broken by the corresponding finite-tail single-unit gadget. Full details are in Appendix A.5. $\qquad\square$

*Remark* 3 (Three layers). Part (a) is what a deployer uses; parts (b) and (c) certify that this bound cannot be tightened at $\mathcal{I}_k$ — globally through the cap, or coordinate-wise through the local weights — without additional information.

*Remark* 4 (The operational certificate is cap-safe in our construction). The cap-active construction is stated for $C_0, C_1$ because the operational certificate $C_2$ does not activate the cap in our construction. Within the constructive class $\mathfrak{M}_{\text{tail}}$ this is immediate: both gadgets realize $C_2$ as the exact realized loss ($C_2 = L$ on the finite-tail gadget, and $C_2 = H\Delta_r = L$ on the cliff-chain), and disjoint composition preserves this, so $C_2 = L \leq R_{\max}$ throughout $\mathfrak{M}_{\text{tail}}$ and the finite-tail gadget realizes $C_2$ exactly. Beyond the constructive class we observe the same numerically: across $6 \times 10^4$ random valid instances (exact dynamic programming under the controller's own occupancy) $C_2 \leq R_{\max}$ without exception. The characterization does not rely on resolving whether $C_2$ can activate the cap for instances outside $\mathfrak{M}_{\text{tail}}$; cap-active sharpness is proved only for $C_0, C_1$, where it occurs.

**Proposition 5** (Non-vacuity of the rich witness class on the cap-inactive subclass). *When $C_k \leq R_{\max}$, Assumption 1 is non-vacuous. Each unit $u = (t, s, i, a_{<i})$ can be realized by a finite-horizon tail gadget of length $H - t$: after the fallback branch, the reference tail earns rewards whose total advantage over the fallback tail is exactly $\omega_k(u)$, distributed over the remaining $H - t$ steps. Since $\omega_k(u) \leq (H - t)\Delta_r$, all rewards remain in $[0, \Delta_r]$, and the additive total loss equals $C_k$.*

**Corollary 3** (Cap-aware necessity of fallback logging). *Fallback logging is necessary to move from $C_1$ to $C_2$ at the information-term level. Globally, under Assumption 1,*

$$\mathcal{C}^{\text{opt}}_{\mathfrak{M}_{\text{R}}}(\mathcal{I}_2) = R_{\max} \wedge C_2 \leq R_{\max} \wedge C_1 = \mathcal{C}^{\text{opt}}_{\mathfrak{M}_{\text{R}}}(\mathcal{I}_1),$$

*with strict improvement if and only if*

$$R_{\max} \wedge C_2 < R_{\max} \wedge C_1.$$

*Equivalently, since $C_2 \leq C_1$, strict improvement holds exactly when $C_2 < C_1$ and $C_2 < R_{\max}$. In the pre-cap coordinate-local sense, $C_2 < C_1$ whenever $W_{\text{fb}} < W$ on a positive-mass set.*

*Proof.* Two instances can share $(\mu, g, W)$ while one fallback distribution places mass on lower-swing actions and the other places mass on worst-swing actions. They are indistinguishable under $\mathcal{I}_1$, so an $\mathcal{I}_1$ certificate must cover the worst case $C_1$. Logging the fallback identity resolves the realized conditional swing $W_{\mathrm{fb}}$, giving $C_2$. The capped strictness condition is exactly the strict inequality between the capped terms. $\square$

**Lemma 2** (Sharpness witnesses). *Each inequality in the pre-cap chain $C_0 \geq C_1 \geq C_2 \geq L$ is tight on an admissible witness: $C_1 = C_2$ when the fallback always selects a worst-swing action; $C_0 = C_1$ in a single-agent single-step all-or-nothing game; and $C_2 = L$ on the cap-inactive finite-tail construction of Proposition 5.*

## 7 Finite-Sample Operational Certificate

We now estimate the operational certificate $C_2$ from fresh certification episodes.

**Definition 10** (Rollout-then-sample protocol). Certification episodes $j = 1, \ldots, m$ are i.i.d. Each episode first rolls out one trajectory under the frozen controller and then samples one coordinate-time unit uniformly among the $nH$ units on that trajectory. This induces $U_j \sim \mu$. Let $F_j$ be the endorsement-failure indicator at $U_j$. If $F_j = 1$, the executed fallback action is logged and a conservative upper bound $W_{\mathrm{fb},j}^+$ on its coordinate disadvantage is obtained by independent resettable rollouts or exact tabular evaluation.

**Assumption 2** (Conservative value bound). The estimated value bounds are conservative: with probability at least $1 - \delta_G$, jointly over all logged failed units,

$$W_{\mathrm{fb},j}^+ \geq \Delta_+(U_j, a^{\mathrm{fb}}).$$

Exact tabular evaluation has $\delta_G = 0$; a generic deep critic does not satisfy this assumption without additional validation. Theorem 4 discharges it by an explicit construction that delivers exactly this high-probability joint event.

**Theorem 3** ($\mathcal{I}_2$ finite-sample certificate). *Each certification episode $j$ draws a unit $U_j \sim \mu$ with failure indicator $F_j$ and, if $F_j = 1$, an independent value-bound estimate $W_{\mathrm{fb},j}^+$ from fresh rollouts at $U_j$, so that the triples $(U_j, F_j, W_{\mathrm{fb},j}^+)$ are i.i.d. across $j$. Let $X_j = nH\, W_{\mathrm{fb},j}^+ F_j$ with deterministic range $0 \leq X_j \leq b_0 = nHB_Q$, where $B_Q$ is a pre-certified value-range bound, and let $\mu_X = \mathbb{E}[X_j]$. Define*

$$\widehat{B}_N = \bar{X} + \sqrt{\frac{2\widehat{\sigma}^2 \ln(2/\delta_B)}{m}} + \frac{7 b_0 \ln(2/\delta_B)}{3(m-1)}.$$

*Under the conservative value bound (Assumption 2, holding with probability at least $1 - \delta_G$ over the fresh-rollout randomness), $\mu_X \geq C_2 \geq L$, and*

$$\mathbb{P}\{L \leq \widehat{B}_N\} \geq 1 - \delta_B - \delta_G,$$

*where $\delta_B$ accounts for the episode sampling and $\delta_G$ for the value-bound construction. The capped certificate $\widehat{B}_N^\wedge = \min\{H\Delta_r, \widehat{B}_N\}$ retains the same guarantee; with exact tabular evaluation $\delta_G = 0$.*

*Proof.* Write $\mathcal{E}_G = \{W_{\mathrm{fb},j}^+ \geq \Delta_+(U_j, a^{\mathrm{fb}})$ for all logged failed units $j\}$ for the joint conservative event; by Assumption 2, $\mathbb{P}(\mathcal{E}_G) \geq 1 - \delta_G$ over the fresh-rollout randomness, with $\delta_G = 0$ under exact tabular evaluation. On $\mathcal{E}_G$ every per-unit value bound dominates the true swing, so

$$\mu_X = \mathbb{E}[nH\, W_{\mathrm{fb},j}^+ F_j] \geq nH\, \mathbb{E}_\mu[gW_{\mathrm{fb}}] = C_2 \geq L$$

by Corollary 1. Separately, the episodes are i.i.d. with $X_1, \ldots, X_m \in [0, b_0]$, so the empirical-Bernstein inequality of Maurer & Pontil (2009) makes the sampling event $\{\mu_X \leq \widehat{B}_N\}$ hold with probability at least $1 - \delta_B$, for any fixed value-bound rule. On $\mathcal{E}_G \cap \{\mu_X \leq \widehat{B}_N\}$ we have $L \leq \mu_X \leq \widehat{B}_N$; since $\mathbb{P}(\mathcal{E}_G^c) \leq \delta_G$ and $\mathbb{P}(\{\mu_X \leq \widehat{B}_N\}^c) \leq \delta_B$, a union bound gives $\mathbb{P}\{L \leq \widehat{B}_N\} \geq 1 - \delta_B - \delta_G$. Since $H\Delta_r \geq L$ deterministically, capping by $\min\{H\Delta_r, \widehat{B}_N\}$ preserves the bound. $\square$

*Remark 5.* The deterministic bound $b_0$ is not the return ceiling $R_{\max}$. It bounds the one-sampled-unit random variable $X_j = nHW_{\mathrm{fb},j}^+ F_j$, whereas $R_{\max}$ caps the final policy value gap.

**Proposition 6** (Rare-unit lower bound)**.** *Fix $nH$ and consider two cap-inactive alternatives that differ only in the fallback swing by $\varepsilon$ at a unit $u^\star$ of occupancy $p$, while sharing the same $\mathcal{I}_1$ profile. Under the rollout-then-sample observation model, any certification procedure that distinguishes the two alternatives with constant probability requires*

$$m = \Omega\left(\frac{1}{p\varepsilon^2}\right).$$

*Proof.* Only episodes landing on $u^\star$ carry information, giving $mp$ effective samples in expectation. Each effective sample has bounded variance and mean shift $\varepsilon$, so the accumulated KL divergence is $O(mp\varepsilon^2)$. Le Cam's two-point method then gives the stated lower bound. Details are in Appendix A.6. $\square$

We now return to the outstanding assumption from Section 7's setup: Theorem 3 required a conservative value bound (Assumption 2) without saying how to obtain one, and the next result supplies an explicit construction.

**Theorem 4** (Conservative rollout value bound)**.** *Fix a value-range bound $B_Q$ and, for each logged failed unit, draw $K$ independent resettable rollouts of the reference tail and $K$ of the fallback-then-reference tail, with returns in $[0, B_Q]$. Let $\widehat{Q}^{\mathrm{ref}}, \widehat{Q}^{\mathrm{fb}}$ be the empirical means and set*

$$W_{\mathrm{fb}}^+(u) = \min\left\{B_Q, \; \left[\widehat{Q}^{\mathrm{ref}}(u, a^{\mathrm{ref}}) - \widehat{Q}^{\mathrm{fb}}(u, a^{\mathrm{fb}})\right]_+ + 2\,\mathrm{rad}(K, \delta')\right\}, \qquad \mathrm{rad}(K, \delta') = B_Q\sqrt{\frac{\ln(2/\delta')}{2K}}.$$

*If $m_F$ units are logged and $\delta' = \delta_G/(2m_F)$, then with probability at least $1 - \delta_G$, jointly over all logged units, $W_{\mathrm{fb}}^+(u) \geq \Delta_+(u, a^{\mathrm{fb}})$. Hence Assumption 2 holds with this construction, and Theorem 3 applies with $\delta_G$ as stated.*

*Proof.* By Hoeffding's inequality, $\mathbb{P}\{\widehat{Q}^{\mathrm{ref}} \geq Q^{\mathrm{ref}} - \mathrm{rad}\} \geq 1 - \delta'$ and $\mathbb{P}\{\widehat{Q}^{\mathrm{fb}} \leq Q^{\mathrm{fb}} + \mathrm{rad}\} \geq 1 - \delta'$ for each logged unit and each tail. On the intersection, $\widehat{Q}^{\mathrm{ref}} - \widehat{Q}^{\mathrm{fb}} \geq (Q^{\mathrm{ref}} - Q^{\mathrm{fb}}) - 2\,\mathrm{rad}$, so $[\widehat{Q}^{\mathrm{ref}} - \widehat{Q}^{\mathrm{fb}}]_+ + 2\,\mathrm{rad} \geq [Q^{\mathrm{ref}} - Q^{\mathrm{fb}}]_+ = \Delta_+(u, a^{\mathrm{fb}})$, using monotonicity of $[\cdot]_+$. A union bound over $m_F$ units and the two tails each (so $2m_F$ events) with $\delta' = \delta_G/(2m_F)$ gives the joint conservative event with probability at least $1 - \delta_G$. Empirical Bernstein may replace Hoeffding to sharpen rad at variance-dependent rates. Here $B_Q$ bounds the range of the tail returns, not the policy value-loss range $R_{\max} = H\Delta_r$. The clip preserves conservativeness: since returns lie in $[0, B_Q]$, the true clipped disadvantage satisfies $\Delta_+(u, a^{\mathrm{fb}}) \leq B_Q$, so whenever the un-clipped bound dominates $\Delta_+(u, a^{\mathrm{fb}})$ so does $\min\{B_Q, \cdot\}$; the clip also guarantees $W_{\mathrm{fb}}^+(u) \leq B_Q$, keeping the sampled variable within the range $b_0 = nHB_Q$ used by Theorem 3. $\square$

The rollout construction therefore requires no exact values: it returns a conservative bound from $K$ resettable rollouts, and the additional conservatism relative to exact-value evaluation shrinks as $O(1/\sqrt{K})$, so exact evaluation of $W$ and $W_{\mathrm{fb}}$ is the $K \to \infty$ limit of a construction that is valid at every finite $K$ rather than a prerequisite for validity.

*Remark* 6 (How Theorems 3 and 4 fit together). The finite-sample certificate has two independent sources of error, and the two theorems handle them separately. Theorem 3 treats the *sampling* error: given any rule that returns a valid conservative upper bound $W_{\mathrm{fb},j}^+$ on each logged unit's swing, the empirical-Bernstein term converts $m$ i.i.d. episodes into a $1 - \delta_B$ bound on the population certificate. Here $b_0 = nHB_Q$ is only the deterministic range of the single-unit variable $X_j$, used by the concentration inequality, and is unrelated to how $W_{\mathrm{fb},j}^+$ is obtained. Theorem 4 then supplies exactly such a rule: it estimates each swing by resettable rollouts and inflates the estimate by a Hoeffding *confidence radius* $\mathrm{rad}(K, \delta')$, so that the conservative event of Assumption 2 holds with probability $1 - \delta_G$. In exact tabular evaluation no rollouts are needed, $W_{\mathrm{fb}}$ is known exactly, and $\delta_G = 0$; the confidence radius is precisely the price of not having exact values. The two failure budgets $\delta_B$ (sampling) and $\delta_G$ (value bound) combine by a union bound into the overall $1 - \delta_B - \delta_G$ guarantee.

*Remark* 7 (Total simulator-call budget). With $m$ certification episodes (one rollout each) and $K$ resettable rollouts for each of the two tails at every failed unit, the expected number of simulated episodes is

$$\mathbb{E}[\mathrm{calls}] = m + 2K\,\mathbb{E}[\#\{j : F_j = 1\}] = m\big(1 + 2K\,\mathbb{E}_{U \sim \mu}[g(U)]\big),$$

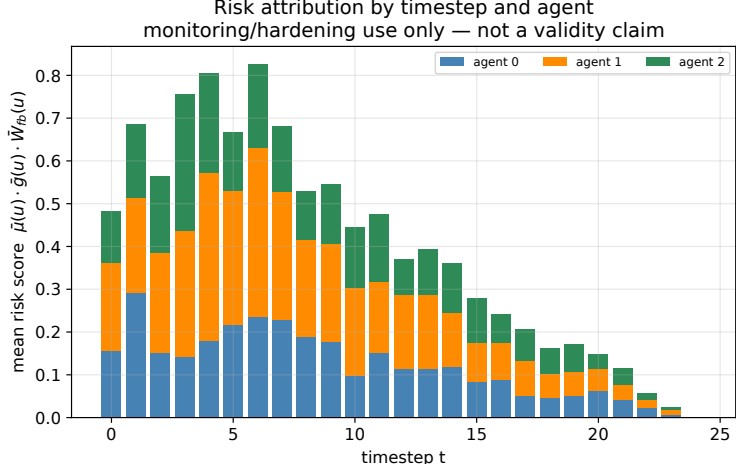

Figure 2: Risk attribution on MPE `simple_spread` (operational, not a validity claim). Per-unit certified-downside score $\bar{\mu}(u)g(u)W_{\text{fb}}(u)$ over 300 deployment episodes, stacked by agent across the horizon. The attributed risk concentrates in the early-to-middle time steps (peak $t \approx 4$–$7$; the first half $t < 12$ carries 74.7%), with no single dominant agent (41%/30%/28%).

and the certificate of Theorem 3 holds with confidence $1 - \delta_B - \delta_G$, where $\delta_G$ is controlled by Theorem 4.

*Remark* 8 (Upper/lower sample complexity match). In the rare-unit regime where a single unit $u^\star$ of occupancy $p$ and fallback-swing shift $\varepsilon$ dominates the variance, the empirical-Bernstein term of Theorem 3 scales as $\sqrt{\hat{\sigma}^2/m}$ with $\hat{\sigma}^2 = \Theta(p\varepsilon^2)$ when the cap is inactive, so resolving an $\varepsilon$-level shift needs $m = \widetilde{\Theta}(1/(p\varepsilon^2))$ episodes. This matches the lower bound $m = \Omega(1/(p\varepsilon^2))$ of Proposition 6 up to logarithmic and constant factors.

## 8 Risk Attribution from the Certificate

The certificate is additive over units:

$$C_2 = \sum_u \bar{\mu}(u)g(u)W_{\text{fb}}(u).$$

Thus $\bar{\mu}(u)g(u)W_{\text{fb}}(u)$ is a risk-attribution score for the certified downside. It identifies state-agent-prefix units that dominate the certificate and can be used for monitoring, review, or hardening. The score attributes the *certified* downside across units, so acting on a high-scoring unit tightens the certificate; whether it lowers the true loss depends on the intervention and is outside the scope of the score.

On the MPE benchmark of Section 9.4, this attribution is operational: computing $\bar{\mu}(u)g(u)W_{\text{fb}}(u)$ over 300 deployment episodes concentrates the certified downside in the early-to-middle horizon (Figure 2) — time steps $t \approx 4$–$7$ carry the peak, and the first half ($t < 12$) accounts for 74.7% of the total attributed risk — spread across agents without a single dominant one (41%/30%/28%), with the top decile of (time-step, agent) units carrying 23.1%. This is a monitoring read, indicating which units to inspect or harden, not a validity claim; here it plausibly reflects the larger ambiguity of landmark assignment before agents commit to targets.

**Corollary 4** (Fixed-occupancy certificate monotonicity)**.** *At fixed occupancy $\mu$ and fixed swings, increasing an odd local committee budget $N_u \mapsto N_u + 2$ on eligible units cannot increase $C_2$. The marginal certificate decrease is*

$$\Delta\widehat{C}_u = \widehat{\bar{\mu}}(u)\,\widehat{W}_{\text{fb}}(u)\big[g_{N_u}(\alpha_u) - g_{N_u+2}(\alpha_u)\big].$$

*The comparison is a certificate-level statement at a fixed occupancy profile; since a larger committee also re-induces the occupancy, it isolates the budget's effect on the certificate and does not by itself imply a change in the true team loss.*

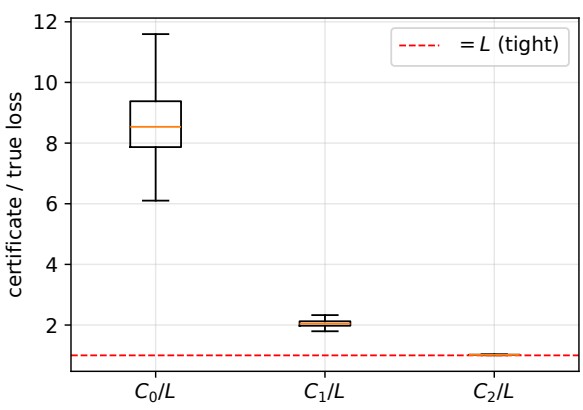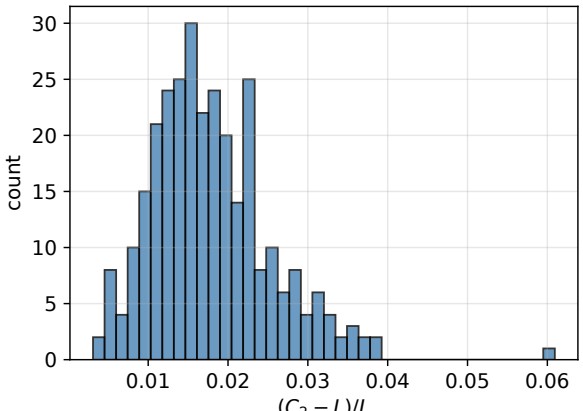

Figure 3: Information chain on 300 exact instances. Left: conservativeness ratios $C_0/L$, $C_1/L$, and $C_2/L$ with medians 8.54, 2.04, and 1.02. Right: slack $(C_2 - L)/L$ of the operational certificate; most slacks concentrate below 0.04, with one visible tail instance. The chain has zero violations.

*Proof.* For eligible units, $g_N(\alpha)$ is non-increasing along odd $N$ by the unanchored version of Lemma 1. The certificate is a non-negative linear form in $g_N$ at fixed occupancy and fixed swings. $\qquad\square$

## 9 Empirical Validation

Our claims are about *validity* — that a certificate is a correct upper bound on the true value-loss — and validity can be checked strictly only where the true loss is known. We therefore evaluate primarily on finite cooperative Markov games small enough for exact dynamic programming, which supplies ground-truth loss and lets every violation rate below be measured against exact truth rather than an estimate. Accordingly the experiments are organized to (i) verify the validity chain and its tightness against exact truth (Section 9.1); (ii) verify finite-sample coverage and the conservative rollout construction that removes the exact-value requirement (Section 9.2); (iii) verify dependence-agnostic behavior under correlated votes (Section 9.3); and (iv) instantiate the certificate on a neural benchmark (Section 9.4). On the neural benchmark the certificate is valid by construction; what that environment lacks is exact ground truth, so its *tightness* is audited only in the tabular settings, not its validity. All headline numbers are recomputed from the per-instance logs in the supplementary material.

### 9.1 Exact validity and information tightness

Across 300 exact cooperative Markov games varying $n$, $H$, odd $N$, endorsement rates, swings, and fallback type, the ordering $C_0 \geq C_1 \geq C_2 \geq L$ holds with zero violations. The median conservativeness ratios are $C_0/L \approx 8.54$, $C_1/L \approx 2.04$, and $C_2/L \approx 1.02$ (Figure 3): the operational certificate is close to the true loss while remaining conservative. The slack distribution concentrates below 0.04 with a single visible tail instance near 0.06, so the tightness is not an artifact of averaging.

Joint occupancy weighting is not a mere technicality. Over 3600 randomized time-inhomogeneous instances, the reference-prefix telescoping bound under-estimates the true loss on a positive fraction (Figure 4), instantiating Proposition 2, while the joint occupancy-weighted certificate $C_2$ stays valid throughout. The marginal-failure closed form of Proposition 1 and the equality-witness sharpness checks are reported in Appendix B.

Beyond the randomized instances, we validate the chain on a family of *structured* cooperative games with genuine coordination semantics: a resource-conflict game in which agents choose whether to claim a shared resource, and a round yields reward only if the number of claimants does not exceed the available capacity, so independently reasonable claims can jointly over-commit and forfeit the reward. Across 315 such games with the number of agents ranging from $n = 2$ to $n = 7$ (Table 3), the ordering $C_0 \geq C_1 \geq C_2 \geq L$ holds with zero

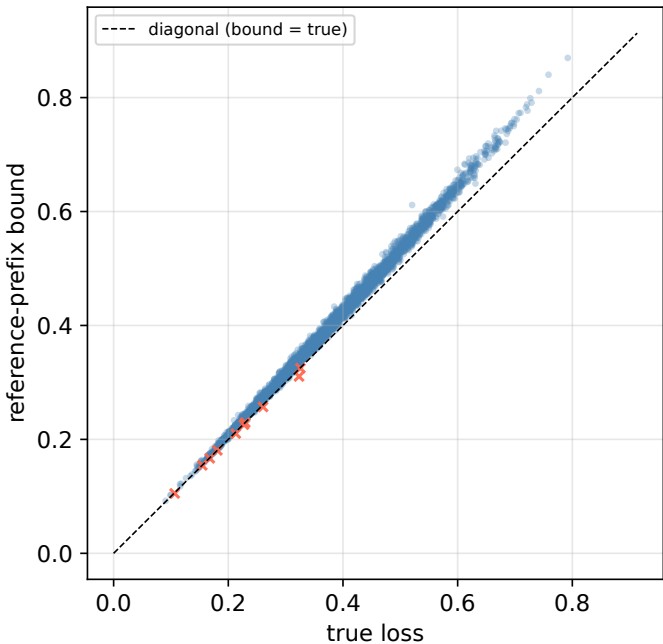

Figure 4: Reference-prefix bound can under-estimate the true loss, over 3600 randomized instances. Any point below the diagonal is a certificate violation; such violations occur even though the joint occupancy-weighted certificate $C_2$ remains conservative. Joint occupancy weighting is necessary for validity.

| $n$ | games | chain viol. | $C_2$ viol. | $C_0/L$ | $C_1/L$ | $C_2/L$ | mean $L$ |
|---|---|---|---|---|---|---|---|
| 2 | 80 | 0 | 0 | 33.98 | 2.00 | 1.005 | 0.19 |
| 3 | 80 | 0 | 0 | 22.42 | 1.69 | 1.015 | 0.44 |
| 4 | 60 | 0 | 0 | 21.07 | 1.65 | 1.015 | 0.71 |
| 5 | 50 | 0 | 0 | 22.83 | 1.67 | 1.028 | 0.88 |
| 6 | 30 | 0 | 0 | 19.78 | 1.59 | 1.015 | 1.27 |
| 7 | 15 | 0 | 0 | 20.24 | 1.59 | 1.017 | 1.44 |

Table 3: Structured resource-conflict cooperative games, $n = 2$ to $7$ (315 games total, exact-DP audited). Zero chain violations and zero $C_2$ violations against exact truth; the operational certificate stays tight ($C_2/L$ medians 1.005–1.028) as the mean true loss grows with $n$, so tightness is not an artifact of small or low-loss games.

violations and the operational certificate stays tight, with per-$n$ median $C_2/L$ between 1.005 and 1.028, while the realized stakes grow with team size (mean true loss $0.19 \to 1.44$). The prefix-drift under-estimation of Proposition 2 persists in this structured family, confirming it is not an artifact of the randomized construction; tightness is maintained even as the games become large and high-stakes, so it does not rely on small or low-loss instances.

## 9.2 Finite-sample and conservative rollout certificates

We instantiate Theorem 3 from $m$ logged episodes and audit against exact-DP truth, with $\delta_G = 0$ in exact tabular evaluation. Coverage is at or above the target confidence level for all tested sample sizes $m \in \{500, 1500, 5000\}$ for both $\mathcal{I}_1$ and $\mathcal{I}_2$ (Figure 5, left), and the logged-fallback estimator is materially tighter (median tightness 1.205 for $\mathcal{I}_2$ versus 2.31 for $\mathcal{I}_1$ at $m = 5000$). Comparing concentration choices at $m = 5000$ on fixed draws (Figure 5, right), empirical Bernstein attains coverage 1.000 at tightness 1.205, Hoeffding 1.000 at 1.431, and clipped empirical Bernstein 0.998 at 1.178; the naive plug-in mean is tightest in

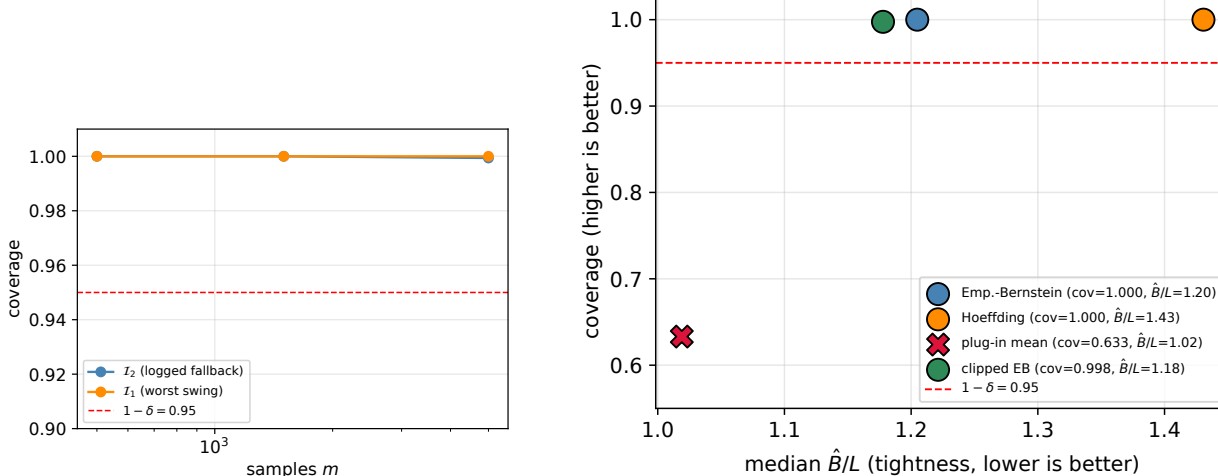

Figure 5: Finite-sample $\mathcal{I}_2$ certificate. Left: empirical coverage of $\mathcal{I}_2$ and $\mathcal{I}_1$ against exact-DP truth at $m \in \{500, 1500, 5000\}$. Right: bound-type comparison at $m = 5000$ on fixed draws — empirical Bernstein (coverage 1.000, tightness 1.205), Hoeffding (1.000, 1.431), clipped empirical Bernstein (0.998, 1.178), naive plug-in (0.633, invalid). Legend values are rounded to two decimals.

ratio (1.019) but under-covers at 0.633 and is therefore invalid. Empirical Bernstein is the operating point as the tightest of the valid choices.

The conservative value-bound construction of Theorem 4 operationalizes Assumption 2. Replacing exact values by conservative resettable-rollout bounds and sweeping the per-tail rollout budget $K \in \{25, 50, 100, 200, 400\}$ on 8 cooperative Markov games (20 repeats each, $m = 2000$ certification episodes, $\delta_B = \delta_G = 0.05$, seed 0), the joint conservative event holds at rate 1.000 for every $K$ (Figure 6, left): validity is unconditional in $K$. The loss-relative tightness improves monotonically with the rollout budget, the median $\widehat{B}_N/L$ falling from 15.36 at $K = 25$ to 5.07 at $K = 400$ (Figure 6, right). Relative to the trivial range cap, the median $\widehat{B}_N/R_{\max}$ falls from 1.53 at $K = 25$ to 0.50 at $K = 400$, dropping below one by $K = 100$ (0.85): with a sufficient rollout budget the conservative certificate is not merely valid but informative in this regime. We stress what this informativeness is and is not: the ratio to the trivial cap $\widehat{B}_N/R_{\max}$ falls below one not because the certificate becomes tight against the true loss — at $K = 400$ it remains roughly 5× the true loss — but because these instances have a small loss-to-range ratio ($L/R_{\max} \approx 0.1$), so the trivial cap $R_{\max}$ is itself loose and a moderately conservative certificate can improve on it. On instances where the loss occupies a larger fraction of the return range, the same rollout budget need not beat the trivial cap; informativeness relative to $R_{\max}$ is therefore a joint property of the rollout budget and the loss-to-range regime, not of the budget alone. What the budget $K$ controls is the conservatism relative to the true loss ($\widehat{B}_N/L$), which shrinks monotonically with $K$; whether that suffices to beat $R_{\max}$ additionally depends on the regime.

### 9.3 Dependence-agnostic certification

Corollary 2 predicts that the certificate built from the failure probability $g$ remains valid for correlated advisors, while a binomial plug-in that assumes i.i.d. votes under-covers. We test this in a controlled tabular construction. At each unit the latent endorsement level is $\Theta_u \sim \text{Beta}(a_u, b_u)$ and the $N = 5$ advisor votes are conditionally i.i.d. Bernoulli($\Theta_u$), i.e. Beta-Binomial($N, a_u, b_u$); the concentration $s = a_u + b_u$ is the sole over-dispersion knob, with the mean endorsement fixed in the $\bar{\alpha} > 1/2$ regime where $g_N$ is convex. The over-dispersion enters *both* the certificate and the exact-DP true loss: the controller's per-unit failure rate is the de Finetti mixture $g(u) = \mathbb{E}_{\Theta_u}[g_N(\Theta_u)]$, so the true loss itself rises as $s$ shrinks ($0.371 \to 0.544$ across the sweep, Figure 7 right).

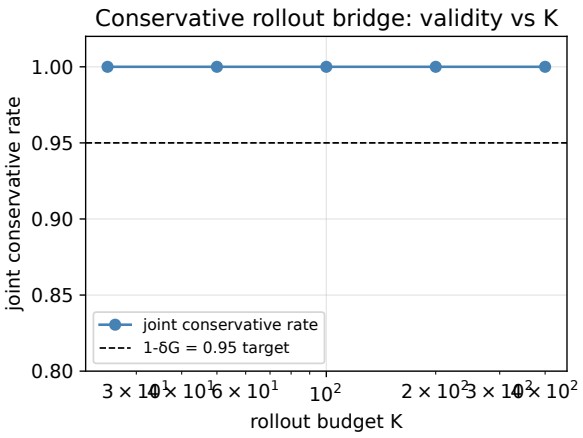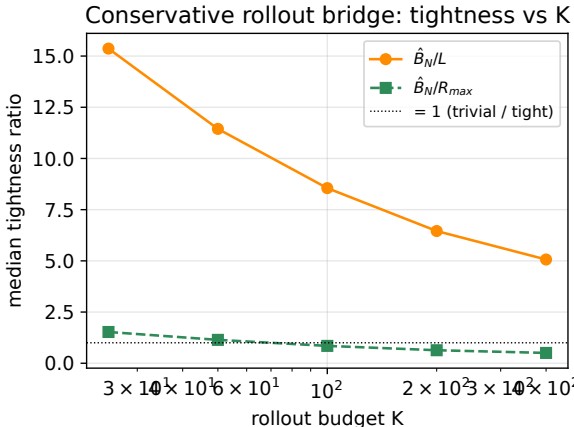

Figure 6: Conservative rollout bridge with the value-bound construction of Theorem 4, audited against exact-DP truth, sweeping the per-tail value-evaluation budget $K$. Left: the joint conservative event holds at rate 1.000 for all $K$. Right: median loss-relative tightness $\widehat{B}_N/L$ falls from 15.36 ($K = 25$) to 5.07 ($K = 400$), and the ratio to the trivial cap $\widehat{B}_N/R_{\max}$ from 1.53 to 0.50; validity is independent of $K$ while additional rollout budget reduces conservatism.

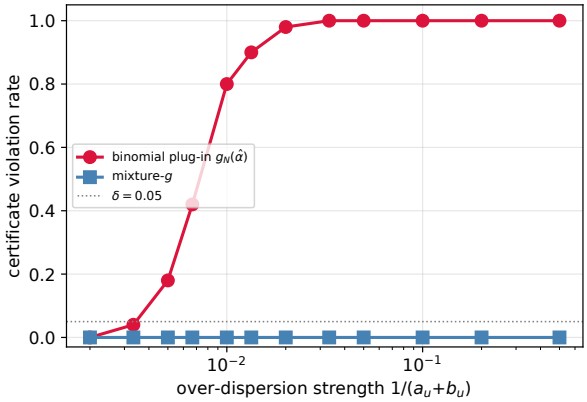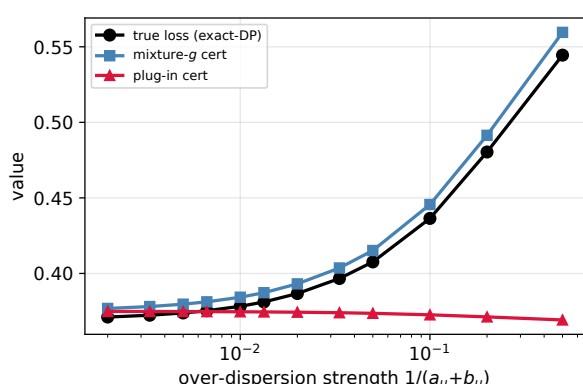

Figure 7: Correlated committees via Beta-Binomial over-dispersion (concentration $s$; smaller $s$ = more over-dispersed). Left: the mixture-$g$ certificate stays valid (coverage 1.000) while the binomial plug-in's under-cover rate rises monotonically to 1.0. Right: the exact-DP true loss itself rises with over-dispersion ($0.371 \rightarrow 0.544$), since the controller's failure rate is the de Finetti mixture.

The mixture-$g$ certificate, which uses the true mixture $g$ (a controlled exact-DP audit), maintains coverage 1.000 with zero violations across all over-dispersion levels (Figure 7, left). The binomial plug-in certificate, which substitutes $g_N(\hat{\alpha}_u)$ at the mean endorsement, under-covers increasingly as concentration shrinks, its under-cover rate rising monotonically from 0.00 at $s = 500$ through $0.18, 0.42, 0.80, 0.90, 0.98$ to 1.00 at $s \leq 30$. The direction is determined by Jensen's inequality on the convex $g_N$, which forces $\mathbb{E}_\Theta[g_N(\Theta)] > g_N(\mathbb{E}[\Theta])$: the plug-in systematically under-estimates the failure rate and hence the loss. This supports the dependence-agnostic certificate as the appropriate validity target under correlated committee votes, and shows that validity does not rely on the binomial form.

## 9.4 Certification on a neural benchmark

The rollout certificate of Section 9.2 requires only a resettable simulator and bounded returns, not exact values, so it applies unchanged to a learned committee in a neural environment. On MPE `simple_spread` we

| Certification budget | Concentration | $\widehat{B}_N/R_{\max}$ | range cap |
|---|---|---|---|
| $m = 150,\ K \in [100, 800]$ | Hoeffding | 14.2–32.2 | active (vacuous) |
| $m = 1500,\ K = 8000$ | empirical Bernstein | 1.008 | active |
| $m = 3000,\ K = 25000$ | empirical Bernstein | **0.494** | inactive (informative) |

Table 4: Rollout certificate on MPE `simple_spread` with a trained IPPO committee ($N = 5$). The swing-only operational quantity is small throughout ($C_2^{\mathrm{emp}} \in [0.07, 0.14]\, R_{\max}$; it is a plug-in estimate, so the small-$m$ value is a noisier under-estimate of the same population quantity). The deployed tightness is set by the finite-sample concentration. At $m{=}3000$, $K{=}25000$ the empirical-Bernstein certificate is informative ($\widehat{B}_N/R_{\max} = 0.494$), with budget-level decomposition (fractions of $R_{\max}$): operational $C_2^{\mathrm{emp}}$ 0.119, empirical-Bernstein radius 0.135, sampling-variance term 0.024, finite-$m$ range term 0.215.

deploy a trained IPPO committee ($N = 5$ independently trained members) as the agreement-gated controller, designating one member (base seed 0) as the reference policy $\pi^{\mathrm{ref}}$ and the other four as advisors, with plurality vote as the fallback rule, and instantiate the certificate of Theorems 3–4 directly from resettable rollouts: the reference tail and the fallback tail are both rolled out under $\pi^{\mathrm{ref}}$, and the committee is used only to determine the endorsement-failure event and the executed fallback action. Because MPE admits no exact-DP ground truth, we report the certificate here but audit its *tightness* only in the tabular settings above.

Two facts make the result interpretable (Table 4). First, the swing-only operational quantity — the plug-in estimate $C_2^{\mathrm{emp}}$ of $C_2$ from the same rollouts, without any confidence radius, used only for diagnosis and not itself a certified bound — is small throughout, $C_2^{\mathrm{emp}} \in [0.07, 0.14]\, R_{\max}$ across budgets, so the certified downside itself is modest and committee disagreement is not the obstacle. Second, the deployed tightness is governed by the finite-sample concentration, and the choice of inequality matters: under the default Hoeffding radius with a small budget ($m = 150$) the certificate is vacuous ($\widehat{B}_N/R_{\max} > 14$), but the tail rollouts are near-deterministic in this environment (the empirical tail standard deviation is on the order of $10^{-6}$ of $B_Q$), so the variance-adaptive empirical-Bernstein radius permitted by Theorem 4 sharpens the bound substantially. With empirical-Bernstein concentration and a larger certification budget ($m = 3000$, $K = 25000$) the certificate reaches $\widehat{B}_N/R_{\max} = 0.494$ with the range cap inactive: a genuinely informative certificate on a standard neural benchmark, obtained with the paper's own constructions and no assumptions beyond the resettable simulator and bounded returns already required by Theorem 4, with the same loss-to-range caveat as in Section 9.2 (here the operational downside is itself small, $C_2^{\mathrm{emp}} \in [0.07, 0.14]\, R_{\max}$). Its budget-level decomposition shows the residual conservatism is dominated by the finite-$m$ range term (Table 4), which the sample budget controls.

## 9.5 Diagnostics and additional checks

Appendix B reports additional checks that are not core to the validity claims: a rare-unit sample-complexity scaling consistent with Proposition 6; a comparison against adjacent off-policy and robust-RL baselines; a computational observation that rollout-based certification avoids exact enumeration under a fixed sampling budget; and the $\eta = 0$ boundary and component-falsification ablations. As a supporting check on the same neural benchmark, the debiased endorsement rate on MPE `simple_spread` with trained IPPO and MAPPO committees (Yu et al., 2022; Rutherford et al., 2024) is stable across committee budgets ($\widehat{p} \approx 0.26$), consistent with the conditional-i.i.d. advisor approximation; the rollout certificate on this benchmark is reported in Section 9.4.

## 10 Discussion, Limitations, and Deployment Scope

The certificate is profile-relative and worst-case over the information available. The validity chain is unconditional, and the optimality claim is established over an explicit constructive witness class rather than an abstract assumption. The carrier is agnostic to the committee's dependence structure: it requires only the failure probability $g$, and the over-dispersion experiment confirms that validity does not rely on the binomial form. The main text uses $\eta = 0$ to isolate endorsement-failure downside; for $\eta > 0$, the success-side term $(1 - g)w_\psi$ is necessary. Logging the fallback identity is low-overhead at deployment, but numerical

evaluation of $W$ and $W_{\text{fb}}$ requires exact values or the conservative confidence bound of Theorem 4. The rollout construction supplies exactly such a bound without a learned critic, and Section 9.4 shows it yields an informative certificate on a neural benchmark once the concentration is variance-adaptive; a generic learned *critic*, by contrast, is not certified unless separately shown to satisfy Assumption 2. Two practical observations follow. On a resettable simulator with near-deterministic tail returns, the Hoeffding radius is needlessly loose and the empirical-Bernstein radius sanctioned by Theorem 4 recovers most of the slack; and once the radius is tightened, the finite-$m$ range term becomes the dominant residual, so the certification budget $m$ is the remaining lever. Whether a value estimator with an explicit confidence penalty, in the spirit of conservative offline value estimation (Kumar et al., 2020), can deliver the uniform conservative guarantee of Assumption 2 without resettable rollouts is left to future work.

### Reproducibility Statement

All experiments are seeded (`seed = 0`); the tabular experiments are CPU-only and the neural-benchmark certificate is computed from resettable MPE `simple_spread` rollouts. We provide anonymized code, configuration files, and per-instance logs in the supplementary material, including a released implementation of the conservative rollout-bridge construction of Theorem 4 (both the Hoeffding and the empirical-Bernstein radius) shared by the tabular and neural-benchmark experiments; all headline numbers are recomputed from these logs at the stated budgets ($m$, $K$, with $\delta_B = \delta_G = 0.05$). Validity experiments are audited against exact dynamic-programming truth; the correlated-committee experiment uses exact-DP truth under the de Finetti mixture failure process; the MPE certificate is reported without a tightness audit, since that environment has no exact ground truth. Per-experiment configurations (instance counts, $m$, $K$, $\delta$, value source) are listed in Appendix B.

## Broader Impact Statement

This work develops deployment-time guarantees for agreement-gated committee controllers in cooperative multi-agent systems. As test-time committees of policies or ensemble advisors are increasingly used to spend additional compute at deployment, the ability to certify their value-loss relative to a reference policy from deployment-time quantities (execution logs together with value evaluation) is directly relevant to the safe operation of such systems. The certificate is conservative by construction and is intended to support, not replace, domain-specific safety review: the certificate bounds the reference-relative downside, but the bound is only as meaningful as the modelling assumptions behind it. Two cautions are worth stating explicitly. First, the strict-validity guarantees hold under exact value evaluation or under the conservative value-bound construction of Theorem 4; a generic learned critic does not satisfy the conservative-value assumption without separate validation, and treating learned-critic outputs as certified could give a false sense of safety. Second, the certificate bounds value-loss against a chosen reference policy and does not by itself adjudicate whether that reference is itself appropriate for the deployment. We therefore recommend that practitioners verify the conservative-value condition and the choice of reference before drawing operational conclusions from any certificate produced by this method.

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

# A Proofs and Additional Details

## A.1 Formal setup

**Definition 11** (Coordinate value). For $u = (t, s, i, a_{<i})$,

$$Q^{\pi^{\mathrm{ref}}}(u, a_i) = \mathbb{E}\left[r_t(s_t, a_t) + V_{t+1}^{\pi^{\mathrm{ref}}}(s_{t+1}) \mid s_t = s, \ a_{<i}, \ a_i, \ a_{>i} \sim \pi^{\mathrm{ref}}\right].$$

The return ceiling follows from bounded rewards: $J_M(\pi^{\mathrm{ref}}) - J_M(\pi) \leq H\Delta_r = R_{\max}$ for every policy $\pi$.

## A.2 Per-agent failure and prefix drift

*Proof of Proposition 1.* The two-agent example gives true loss

$$2g(1 - g)(0.1) + g^2 = 0.2g(1 - g) + g^2,$$

while the marginal certificate gives $0.1g + 0.1g = 0.2g$. For $g = 0.317$, the true loss is approximately 0.1438 and the marginal certificate is approximately 0.0634. For the $n$-agent quadratic family, if $K \sim \mathrm{Bin}(n, g)$ is the number of deviating coordinates, then

$$\mathbb{E}\left[\left(\frac{K}{n}\right)^2\right] = \frac{ng(1 - g) + (ng)^2}{n^2} = \frac{g(1 - g)}{n} + g^2.$$

The marginal certificate is $g/n$, so the ratio of true loss to marginal certificate is $(1 - g) + ng$. $\square$

*Proof of Proposition 2.* The reference-prefix bound evaluates coordinate swings only at $a_{<i}^{\mathrm{ref}}$, while the deployed controller visits non-reference prefixes with positive probability. The joint certificate integrates over the deployed prefix law:

$$C_2 = \sum_{t,s,i,a_{<i}} d_t^{\pi^{\mathrm{ctrl}}}(s)\mathbb{P}(a_{<i} \mid t, s)g(t, s, i, a_{<i})W_{\mathrm{fb}}(t, s, i, a_{<i}).$$

In the constructed time-inhomogeneous instance, exact dynamic programming gives true loss 0.21290, reference-prefix bound 0.20947, and joint certificate 0.21489, proving strict invalidity of the reference-prefix certificate. $\square$

### A.3 Carrier bound

*Proof of Theorem 1.* For finite-horizon Markov games, the performance-difference identity gives

$$J(\pi^{\text{ref}}) - J(\pi^{\text{ctrl}}) = \sum_{t=0}^{H-1} \mathbb{E}_{s_t \sim d_t^{\pi^{\text{ctrl}}}} \left[ V_t^{\pi^{\text{ref}}}(s_t) - \mathbb{E}_{a_t \sim \pi_t^{\text{ctrl}}(\cdot|s_t)} Q_t^{\pi^{\text{ref}}}(s_t, a_t) \right].$$

Since $V_t^{\pi^{\text{ref}}}(s) = \mathbb{E}_{a \sim \pi_t^{\text{ref}}(\cdot|s)} Q_t^{\pi^{\text{ref}}}(s, a)$, decompose the joint-action difference by replacing the coordinates in a fixed virtual order. Conditioning on the deployed prefix $a_{<i}$, the $i$th increment is

$$Q^{\pi^{\text{ref}}}(u, a_i^{\text{ref}}) - \mathbb{E}_{a_i \sim \pi^{\text{ctrl}}(\cdot|u)} Q^{\pi^{\text{ref}}}(u, a_i) \leq \mathbb{E}_{a_i \sim \pi^{\text{ctrl}}(\cdot|u)} \Delta_+(u, a_i),$$

because

$$Q^{\pi^{\text{ref}}}(u, a_i^{\text{ref}}) - Q^{\pi^{\text{ref}}}(u, a_i) \leq \left[ Q^{\pi^{\text{ref}}}(u, a_i^{\text{ref}}) - Q^{\pi^{\text{ref}}}(u, a_i) \right]_+.$$

At unit $u$, the controller succeeds with probability $1 - g(u)$ and incurs at most $w_\psi(u)$; it fails with probability $g(u)$ and incurs conditional mean $W_{\text{fb}}(u)$. Therefore the coordinate increment is at most $(1 - g(u))w_\psi(u) + g(u)W_{\text{fb}}(u)$. Summing over $t$, states, coordinates, and prefixes yields

$$J(\pi^{\text{ref}}) - J(\pi_N^{\text{ctrl}}) \leq \sum_u \bar{\mu}(u) \left[ (1 - g(u))w_\psi(u) + g(u)W_{\text{fb}}(u) \right] = nH \mathbb{E}_{U \sim \mu}[(1 - g)w_\psi + gW_{\text{fb}}].$$

$\square$

### A.4 Success-side witness for Proposition 4

Define the two success-side information levels $\mathcal{I}_\eta^{\text{succ}} = \sigma(\mu, g, \eta)$ and $\mathcal{I}_\psi^{\text{succ}} = \sigma(\mu, g, \eta, w_\psi)$, with $\mathcal{I}_\eta^{\text{succ}} \preceq \mathcal{I}_\psi^{\text{succ}}$. We show that $S_\eta \geq S_\psi$ is attained, so that no $\mathcal{I}_\eta^{\text{succ}}$-measurable certificate can undercut $S_\eta$ while remaining valid.

Fix a one-step instance ($H = 1$) with $g = 0$. Two consequences follow. First, the loss is entirely success-side, $L = nH \, w_\psi$, since no failure occurs. Second, with no continuation the unit occupancy $\mu$ does not depend on the executed action, so changing the endorsed representative's reward gap leaves $\mu$ unchanged. Consider two such instances: instance $\mathcal{A}$ executes an endorsed representative of clipped disadvantage $\Delta_+ = \eta$ (the worst admissible endorsed action), and instance $\mathcal{B}$ one of disadvantage $\Delta_+ = 0$ (an endorsed action of reference value). Both are legal, $L = nH \, w_\psi \leq nH\eta \leq R_{\max}$, and both present the identical $\mathcal{I}_\eta^{\text{succ}}$-profile $(\mu, g=0, \eta)$, yet their success-side losses are $nH\eta$ and $0$.

A certificate measurable in $\mathcal{I}_\eta^{\text{succ}}$ is constant on this shared profile; to remain valid on $\mathcal{A}$ it must be at least $nH\eta = S_\eta$. Logging the executed representative's identity raises the information to $\mathcal{I}_\psi^{\text{succ}}$, resolves $w_\psi$, and yields $S_\psi$. Hence $S_\eta \geq S_\psi$, with equality iff $w_\psi \equiv \eta$ on the support. The tightening mechanism mirrors the failure-side $C_1 \to C_2$: a worst-case charge over an admissible set is replaced by the realized charge once the executed action's identity is observed.

### A.5 Information characterization

**Lemma 3** (Admissible units are locally cap-inactive). *For every admissible unit law and every information level $k$,*

$$\bar{\mu}(u)g(u)\omega_k(u) \leq R_{\max}.$$

*Proof.* By construction, $\bar{\mu}(u) = d_t^{\pi^{\text{ctrl}}}(s)\mathbb{P}(a_{<i} \mid t, s) \leq 1$, $g(u) \leq 1$, and $\omega_k(u) \leq (H - t_u)\Delta_r \leq H\Delta_r = R_{\max}$. Multiplying the three inequalities proves the claim. $\square$

**Lemma 4** (Finite-tail unit gadget). *For any unit $u = (t, s, i, a_{<i})$ and any $\omega(u) \in [0, (H - t)\Delta_r]$, there is a deterministic tail gadget with rewards in $[0, \Delta_r]$ whose reference-minus-fallback value gap equals $\omega(u)$.*

*Proof.* Let $\omega(u) = q\Delta_r + r$ with integer $q \in \{0, \dots, H - t\}$ and $r \in [0, \Delta_r)$, truncating $q$ at $H - t$ with $r = 0$ when equality holds. Along the reference tail, assign reward $\Delta_r$ for $q$ future steps and reward $r$ for one additional step if $r > 0$, with zero elsewhere. Along the fallback tail, assign zero on those steps and match the reference rewards elsewhere. The total gap is $q\Delta_r + r = \omega(u)$ and every reward lies in $[0, \Delta_r]$. $\qquad\square$

**Lemma 5** (Cliff-chain cap-active gadget). *For every $H \geq 2$ and $\Delta_r > 0$ there is a single-agent ($n = 1$) deterministic instance with rewards in $[0, \Delta_r]$ and an agreement-gated controller, realizing the information profile $\bar{\mu}(u_t) = 1$, $g(u_t) = 1$, $W(u_t) = (H - t)\Delta_r$, $W_{\mathrm{fb}}(u_t) = \Delta_r$ at the units $u_t = (t, s_t, 1, \varnothing)$, $t = 0, \dots, H - 1$, whose true loss is exactly $R_{\max} = H\Delta_r$ while*

$$C_0 = C_1 = \frac{H(H+1)}{2}\Delta_r > R_{\max}.$$

*Hence $R_{\max} \wedge C_k = R_{\max} = L$ for $k \in \{0, 1\}$: the cap-active optimum is attained constructively.*

*Proof.* Take states $s_0 \to s_1 \to \cdots \to s_{H-1}$ on a chain plus an absorbing dead state $D$ with reward 0. The reference policy takes, at each $s_t$, an action earning $\Delta_r$ and moving to $s_{t+1}$, so $V^{\pi^{\mathrm{ref}}}(s_t) = (H - t)\Delta_r$ and $V^{\pi^{\mathrm{ref}}}(D) = 0$. At each $s_t$ the controller fails ($g = 1$, e.g. endorsement rate 0) and executes the fallback $a^{\mathrm{fb}}$ that earns 0 and moves to $s_{t+1}$; thus the controller earns 0 throughout and $L = H\Delta_r - 0 = R_{\max}$. The available actions at $s_t$ include a cliff action earning 0 and moving to $D$. Then $Q^{\pi^{\mathrm{ref}}}(u_t, a^{\mathrm{ref}}) = (H - t)\Delta_r$, $Q^{\pi^{\mathrm{ref}}}(u_t, a^{\mathrm{fb}}) = (H - 1 - t)\Delta_r$, and $Q^{\pi^{\mathrm{ref}}}(u_t, \mathrm{cliff}) = 0$, so $W(u_t) = (H - t)\Delta_r$ and $W_{\mathrm{fb}}(u_t) = \Delta_r$. Occupancy is deterministic, $\bar{\mu}(u_t) = 1$. Summing, $C_0 = C_1 = \sum_{t=0}^{H-1}(H - t)\Delta_r = \frac{1}{2}H(H + 1)\Delta_r > H\Delta_r = R_{\max}$ for $H \geq 2$, while $C_2 = \sum_{t=0}^{H-1}\Delta_r = H\Delta_r = R_{\max}$. All rewards lie in $[0, \Delta_r]$. The profile is $\mathcal{I}_1$-consistent with the stated certificate values, and the true loss equals $R_{\max} = R_{\max} \wedge C_k$ for $k \in \{0, 1\}$. $\qquad\square$

*Proof of Proposition 5.* For each unit, use Lemma 4 to realize the desired coordinate gap $\omega_k(u)$ over the remaining horizon. Assign the unit occupancy and failure probability according to the admissible profile. Since $C_k \leq R_{\max}$, summing the unit-level losses does not exceed the feasible return range. The total loss is exactly

$$\sum_u \bar{\mu}(u)g(u)\omega_k(u) = C_k,$$

and the rewards remain bounded by the tail-gadget construction. $\qquad\square$

*Proof of Theorem 2.* Part (a) follows from Corollary 1 and $W_{\mathrm{fb}} \leq W \leq (H - t)\Delta_r$, plus $L \leq R_{\max}$. For part (b), part (a) gives $\mathcal{C}^{\mathrm{opt}}_{\mathfrak{M}_{\mathrm{tail}}}(\mathcal{I}_k) \leq R_{\max} \wedge C_k$. The cap-inactive case $C_k \leq R_{\max}$ is attained by the finite-tail gadget (Lemma 4, via Proposition 5) for all $k$. The cap-active case $C_k > R_{\max}$ is attained for $k \in \{0, 1\}$ by the cliff-chain gadget (Lemma 5), which realizes true loss $R_{\max}$ with $C_0 = C_1 > R_{\max}$, so $R_{\max} \wedge C_k = R_{\max} = L$. Hence equality holds over $\mathfrak{M}_{\mathrm{tail}}$, and over any $\mathfrak{M}_{\mathrm{R}} \supseteq \mathfrak{M}_{\mathrm{tail}}$. For part (c), Lemma 3 allows a single-unit finite-tail witness for every positive-mass unit. Any coordinate-local pre-cap certificate that reduces the pointwise weight below $\omega_k(u)$ on such a set is violated by that witness. $\qquad\square$

*Proof of Lemma 2.* The equality $C_1 = C_2$ holds when the logged fallback always selects an action attaining $W$. The equality $C_0 = C_1$ holds in a single-step all-or-nothing game where $W = (H - t)\Delta_r = \Delta_r$. The equality $C_2 = L$ holds for the cap-inactive finite-tail witness of Proposition 5. $\qquad\square$

## A.6 Finite-sample certificate and lower bound

*Proof of Proposition 6.* Let $P_0$ and $P_1$ differ only at unit $u^\star$, whose occupancy under the rollout-then-sample protocol is $p$. Conditional on landing on $u^\star$, the observation distribution has means separated by $\varepsilon$ and bounded variance, so the one-hit KL is $O(\varepsilon^2)$. The total KL over $m$ episodes is $O(mp\varepsilon^2)$. If $m \lesssim 1/(p\varepsilon^2)$, the total variation distance remains bounded away from one by Pinsker's inequality, and Le Cam's method implies constant error for any test. Thus constant-probability distinction requires $m = \Omega(1/(p\varepsilon^2))$. $\qquad\square$

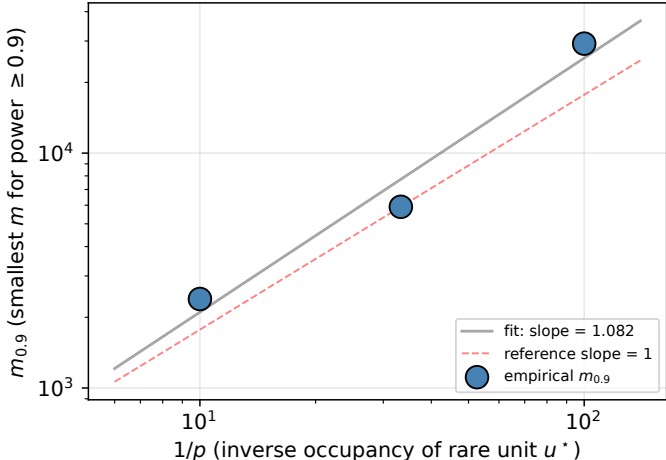

Figure 8: Rare-unit scaling. Episodes $m$ to detect an $\varepsilon$-shift at a unit of occupancy $p$, against $1/p$ on log-log axes. The fitted slope is $1.082$ ($R^2 = 0.966$), consistent with $m = \Theta(1/p)$.

### A.7 Supporting lemmas

*Proof of Lemma 1.* Write $M = N - 1$, $k = (N - 3)/2$, and $q = 1 - \alpha$. Let $b_M(r) = \binom{M}{r}\alpha^r q^{M-r}$. A two-step binomial-CDF identity gives

$$h_{N+2}(\alpha) - h_N(\alpha) = q^2 b_M(k+1) - \alpha^2 b_M(k).$$

Since

$$\frac{b_M(k+1)}{b_M(k)} = \frac{M-k}{k+1}\frac{\alpha}{q},$$

the increment is negative iff

$$\frac{q}{\alpha}\frac{M-k}{k+1} < 1.$$

Substituting $M - k = (N + 1)/2$ and $k + 1 = (N - 1)/2$ yields $\alpha > (N + 1)/(2N)$, with equality at the threshold. $\square$

## B  Additional Experiments

**Rare-unit sample-complexity scaling.**  As a sanity check consistent with the rare-unit lower bound (Proposition 6), we measure with pure exact-DP evaluation the episode count $m$ needed to detect an $\varepsilon$-level fallback-swing shift at a unit of occupancy $p$. Over the feasible range $p \in \{10^{-1}, 3 \times 10^{-2}, 10^{-2}\}$ the required $m$ is $\{2393, 5907, 29240\}$, a log-log slope of $1.082$ against $1/p$ with $R^2 = 0.966$ (Figure 8), consistent with the $m = \Theta(1/p)$ scaling of Proposition 6 and Remark 8. For $p \leq 3 \times 10^{-3}$ the theoretical sample size exceeds $10^6$ and is computationally infeasible to verify directly; those points are archived rather than reported.

**Adjacent baselines from OPE and robust RL.**  We calibrate the $\mathcal{I}_2$ certificate against adjacent off-policy and robust-RL alternatives (8 games, 2000 rollouts each), auditing all against exact truth (Table 5). The $\mathcal{I}_2$ certificate attains coverage $1.000$ at median bound/loss $1.019$. Per-decision importance sampling (PDIS) and the doubly-robust estimator are also valid but substantially looser, and the robust-simulation-lemma bound is valid but uninformative. The comparison is intended to calibrate the certificate against adjacent alternatives rather than to claim dominance over OPE methods in their native estimation setting: the relevant structural difference is that OPE baselines require importance weights and do not produce the per-unit attribution of Section 8.

| Method | Coverage | Median bound/loss | Comment |
|---|---|---|---|
| $\mathcal{I}_2$ certificate (this paper) | 1.000 | 1.019 | tightest valid |
| PDIS (Thomas et al., 2015) | 1.000 | 1.607 | valid, $\approx 58\%$ looser |
| Doubly-robust (Jiang & Li, 2016) | 1.000 | 1.619 | valid, $\approx 58\%$ looser |
| Robust simulation lemma (Kakade & Langford, 2002) | 1.000 | 4.438 | valid but uninformative |

Table 5: Adjacent OPE / robust-RL baselines versus the $\mathcal{I}_2$ certificate, audited against exact truth (8 games, 2000 rollouts each). All are valid; the $\mathcal{I}_2$ certificate is the tightest, and the structural difference rather than the numerical gap is the point.

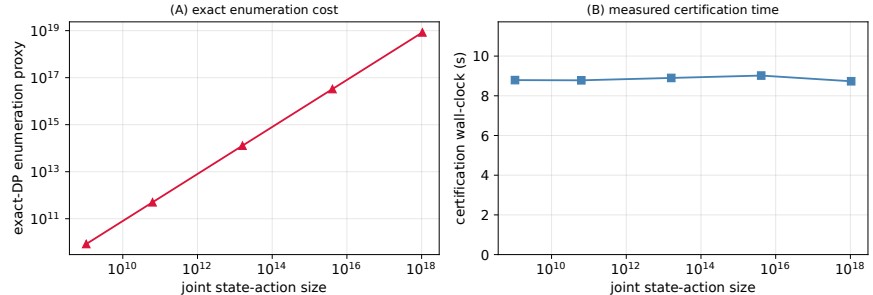

Figure 9: Computational observation on enumeration. The left panel reports an exact-DP enumeration proxy; the right panel reports the measured wall-clock of the rollout certificate under a fixed rollout budget. The figure supports infeasibility of exact enumeration at large joint state-action sizes, not strict validity on realistic neural MARL.

**Computational observation on enumeration.** Fixing $n = 4$, $H = 8$, and $m = 300$ and varying the grid size, the rollout certificate uses a fixed number of sampled units while exact DP would require enumeration over the joint state-action space. Under a fixed sampling budget the measured certification wall-clock stays nearly flat (Figure 9); the exact-DP curve is an enumeration proxy, not a measured wall-clock in the same unit. This is a computational observation that rollout-based certification avoids exact enumeration under fixed sampling budgets, not a validity claim for realistic neural MARL.

$\eta = 0$ **boundary.** At $\eta = 0$, the value-loss bound is never violated across 1200 instances. For $\eta \geq 0.1$, the pure failure term $nH\mathbb{E}[gW_{\mathrm{fb}}]$ that drops the success-side contribution is violated in every instance (Figure 10).

**Falsification of certificate components.** Replacing the operational certificate $C_2$ by ablated variants and measuring against exact truth (Figure 11) shows that $C_2$ itself is never violated, the unweighted variant is violated in 99.7% of instances, the plug-in-$\alpha$ variant in 19.3%, and the no-logging variant $C_1$ remains valid but pays a median slack of 0.43 above $C_2$.

**Rank consistency and budget monotonicity.** The fixed-occupancy committee-sizing checks show certificate monotonicity in the budget in all tested instances, Spearman rank correlation 1.0 between certificate and true loss with no reversals (reversal fraction 0.000), and no true-loss increase in 99.3% of tested fixed-occupancy budget increases. Because this is a diagnostic use of the certificate rather than a main contribution, we summarize it numerically here and leave the budget-drift plots out of the paper.

**Sharpness witnesses.** Closed-form witness values match exact-DP computations up to numerical precision; the maximum occupancy reconstruction error across 85 witnesses is $3.3 \times 10^{-16}$, confirming the equality cases of Lemma 2.

**Per-experiment configurations.** Table 6 lists the configuration of each experiment, including instance counts, sample sizes, and value source; all finite-sample runs use $\delta_B = \delta_G = 0.05$.

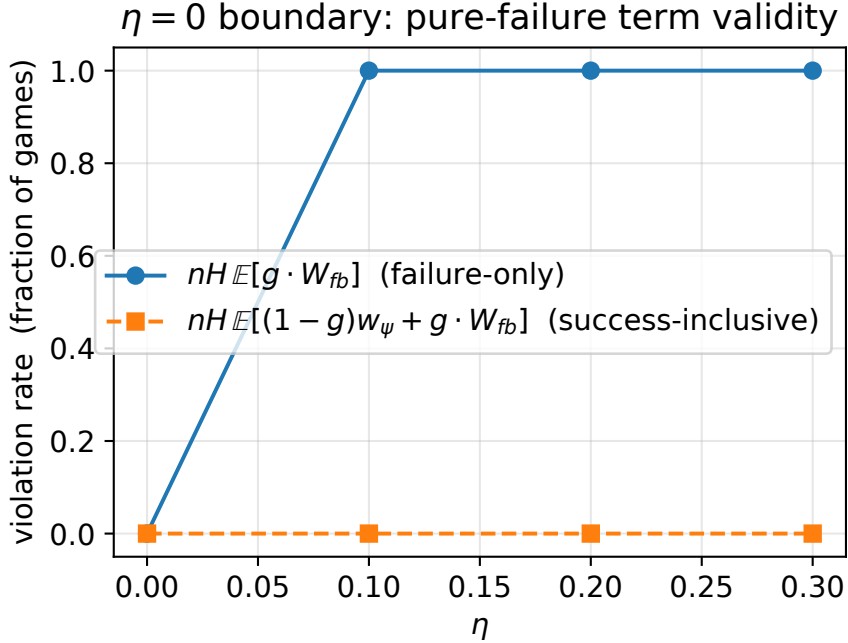

Figure 10: $\eta = 0$ boundary. Violation rate of the pure-failure term as a function of $\eta$: zero at $\eta = 0$ and one for $\eta \geq 0.1$, confirming that the success-side term is not removable for $\eta > 0$.

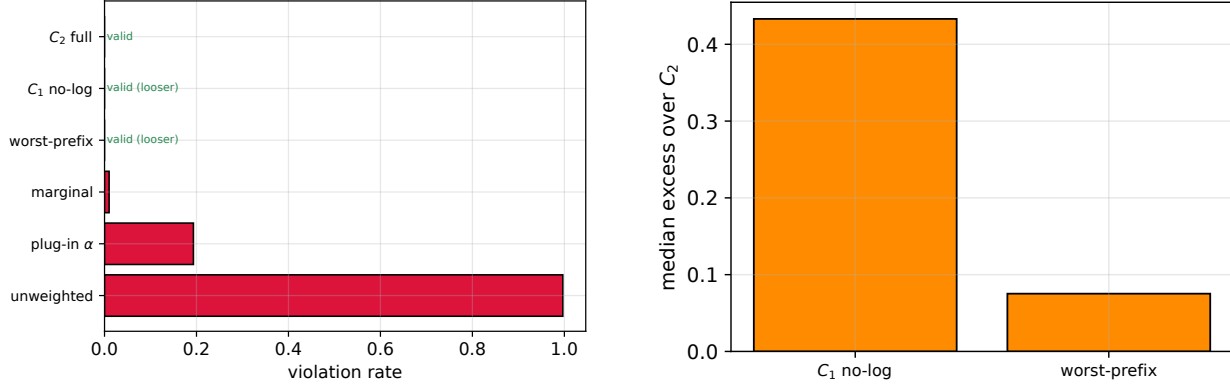

Figure 11: Falsification of certificate components. Left: the full logged-fallback certificate $C_2$ and conservative no-log variants remain valid, while removing occupancy weighting or calibrated failure probabilities breaks validity. Right: the no-log variants are valid but looser than $C_2$; they are not failed baselines.

| Experiment | Instances | $m$ / repeats | Value source |
|---|---|---|---|
| Exact-tabular chain | 300 games | — | exact DP ($\delta_G$=0) |
| Finite-sample / bound comp. | 8 games | $m \in \{500, 1500, 5000\}$, 200 rep | exact DP |
| K-sweep (Thm 4) | 8 games | 20 rep, $m$=2000, $K \in \{25, \ldots, 400\}$ | cons. rollout |
| Rare-unit scaling | $p \in \{10^{-1}, 3 \times 10^{-2}, 10^{-2}\}$ | — | exact DP |
| Adjacent baselines | 8 games | 2000 rollouts | exact truth |
| Correlated committees | 50 games $\times$ 12 levels | $N$=5, $s \in [2, 500]$ | exact DP (mixture) |
| Structured games (Table 3) | 315 games, $n \in [2, 7]$ | — | exact DP |
| MPE certificate (Table 4) | `simple_spread`, $N$=5 IPPO | $m \in \{150, 1500, 3000\}$, $K \leq 25000$ | resettable rollout |

Table 6: Per-experiment configurations. All runs use seed 0 and are reproducible from the per-instance logs in the supplementary material.

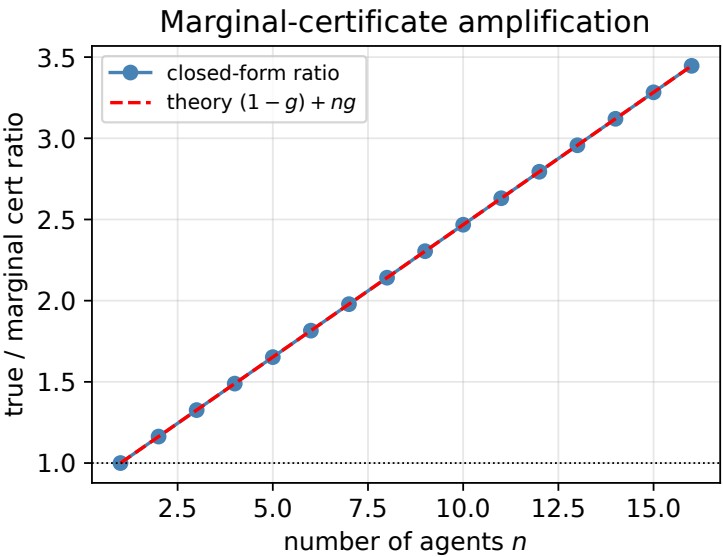

Figure 12: Marginal-certificate failure and amplification (closed-form check for Proposition 1). On the canonical $n$-agent family, the ratio of true team value-loss to the per-agent marginal certificate follows the predicted $(1-g)+ng$ law and equals 1 at $n=1$, so the per-agent certificate under-estimates the true loss for $n \geq 2$.

