# OpenReview forum: "Information-Tight Value-Loss Guarantees for Test-Time Committees in Cooperative MARL"
_TMLR — Under review for TMLR_

### Review · Reviewer_7MPk · 2026-07-03

**Summary Of Contributions:**

The paper studies the certification of value loss for a frozen test-time committee controller in cooperative Multi-Agent Reinforcement Learning (MARL). The main idea is to compare the committee controller against a fixed reference policy $(\pi^{ref})$ and to upper-bound the reference-relative loss using deployment quantities such as controller occupancy, endorsement failure probability, worst-coordinate swing, and logged fallback-action swing. The proposed hierarchy $C_0 \geq C_1 \geq C_2$ is a useful way to show how additional logging can tighten the certificate. The core idea of the paper is to claim that valid certification must be based on the deployed controller’s occupancy rather than the reference policy’s occupancy, which is really good. However, the paper is very difficult to read in its current form. The Introduction has no citations, the definitions and theorem statements are not well-motivated, and many of the notations are not explained in the appropriate places.

**Additional Comments:**

The paper has a potentially strong idea. However, the current manuscript is not yet sufficiently clear or well-motivated for acceptance. The main issues are presentation, missing citations in the Introduction, weak motivation for definitions and theorems, and insufficiently prominent discussion of the value-evaluation assumptions.

**Audience:**

Yes

**Audience Explanation:**

Researchers working on safe reinforcement learning, cooperative MARL, test-time compute, ensembles, or policy certification would likely find the problem interesting. The paper’s main idea, certifying the downside of a fixed committee controller rather than only reporting empirical committee gains, is relevant.

The deployed-occupancy argument is especially interesting. It correctly highlights that using the reference policy’s state distribution can produce invalid certificates in sequential settings. However, the paper’s current readability issues make it difficult for the contribution to come through.

**Broader Impact Concerns:**

I do not see major, unaddressed broader-impact concerns.

**Claims And Evidence:**

No

**Claims Explanation:**

The core theoretical idea appears good, and the paper provides several useful counterexamples and synthetic/tabular experiments. However, the current presentation does not make the claims sufficiently clear or convincing. In particular, the Introduction makes broad claims about cooperative MARL, test-time committees, certification, and deployment-time compute without citing the relevant literature. This is a significant weakness because these are established research areas. Therefore, proper motivation with citations must be clearly written. The definitions are also introduced too abruptly. Important objects such as units, prefixes, $g$, $\mu$, $W$, $W\_{fb}$, and the information sets $I\_0,I\_1,I\_2$ are not sufficiently motivated before being used. The theorem statements are similarly hard to parse because validity, tightness, sharpness, and finite-sample certification are all mixed together.

I am also not fully convinced by the “deployment-time observable” framing. While failure indicators and fallback identities are observable, the quantities $W$ and $W_{fb}$ require evaluation of their exact values. This is a strong assumption and should be stated more prominently.

The main clean certificate applies to the restrictive $\eta=0$ regime, where successful endorsement executes the reference action. This is a narrow setting for a test-time committee. For $\eta>0$, the paper needs an additional success-side term, and the appendix shows that dropping this term leads to violations. This weakens the central simple message of the paper.

The controller model is confusing. The setup first assumes conditionally i.i.d. advisors, but later emphasizes dependence-agnostic validity for arbitrarily correlated advisors.

The paper is unclear regarding finite-sample theory. Theorem 3 assumes a deterministic range bound for $X\_j$, but Theorem 4 adds a confidence radius to the estimated fallback disadvantage. Authors should clearly mention these things.

The empirical evidence is very weak. Most validity checks are synthetic or tabular and audited against exact dynamic-programming truth. The more realistic learned-committee experiment on MPE is explicitly described as diagnostic only. Therefore, the experiments do not convincingly demonstrate that the method is useful for realistic cooperative MARL deployments.

**Requested Changes:**

1.  Add citations in the Introduction. The paper should immediately position itself relative to cooperative MARL, safe RL, off-policy evaluation, performance-difference lemmas, and test-time committees.

2.  Rewrite the setup section with more motivation. Explain why prefix-aware units are needed before introducing $u=(t,s,i,a_{<i})$. Explain the intuition behind $g, W, W_{fb}$, and the information hierarchy before giving formal definitions.

3.  Add a notation table. The current notation is too dense, making the paper unnecessarily hard to follow.

4. The paper did not discuss whether the theory assumes finite state and action spaces, which is the first basic thing to state while motivating the problem.

4.  Separate the main claims more clearly with motivations.

5. In the controller model, define the general failure-probability-based controller first and present the i.i.d. binomial model only as a special case.

Overall, this paper is poorly written and very hard to understand because of its poor presentation. Apart from the above changes, the paper needs a complete revision, as most of the content is very difficult to understand due to poor presentation and the haphazard use of numerous notations without explanations in the right places.

---

> ### Author Response · Authors · 2026-07-06
> **presentation, citations, and notation**
>
> We have revised the manuscript to address the review's comments and the six Requested Changes
>
> **Citations and positioning in the Introduction.** The Introduction cites the test-time-compute literature (repeated sampling, voting, debate, verifier-based aggregation) and the cooperative-MARL formalisms (Markov games, Dec-POMDPs) at first mention, rather than only in Related Work. It also positions the certificate against three neighboring guarantee types the review named. In a new paragraph, "What kind of guarantee this is," we distinguish the certificate from safe and conservative policy improvement, whose object is a policy being selected or improved; from off-policy evaluation, which estimates a target policy's value from logged data via importance weights; and from the performance-difference lemma itself -- the vehicle for our carrier bound, not the contribution. Our controller, by contrast, is frozen -- neither trained nor selected. The certificate simply upper-bounds its reference-relative downside from deployment-time information, meaning logs together with the value-evaluated coordinate gaps described in the next comment. Related Work now follows this same logic, organized by methodological boundary through a comparison table (Table 1) rather than as a chronological list.
>
> **Motivation for the definitions and the notation table.** Section 3 (Problem Setup) now opens with three short paragraphs of motivation before any formal definition. "Units and prefixes" explains the unit $u=(t,s,i,a_{<i})$ before defining it: the executed prefix matters because earlier coordinates may already differ from the reference, and that is exactly what the counterexamples in Section 4 exploit. The next two paragraphs do similar work for the remaining machinery -- "Swings and the failure probability" gives the intuition behind $g$, $W$, and $W_{fb}$ before Definitions 2 and 3 formalize them, and "The information hierarchy" explains in plain language why knowing more tightens the certificate from $C_0$ to $C_1$ to $C_2$, ahead of Theorem 2's formal chain. A notation table (Table 2) follows immediately. Definition 1, meanwhile, now states as its first content that the state space, the per-agent action spaces, and the horizon are all finite -- the point Requested Change 4 asked for -- and notes that Theorem 4's learned-setting extension is what relaxes this assumption.
>
> **Separating the different claim types bundled inside the main theorem.** Theorem 2 used to bundle validity, optimality, and sharpness into three parts of one theorem, without much guidance on what each part actually claims. A short paragraph now precedes it: part (a) is the deployable validity guarantee, requiring only bounded rewards; part (b) is a profile-relative optimality statement over an explicit constructive witness class, not an abstract existence claim. Two things are deliberately kept out of Theorem 2 itself: part (c) -- pre-cap coordinate-local sharpness, a genuinely different kind of property from parts (a) and (b) -- and finite-sample estimation of the operational certificate, which is handled separately in Section 7.
>
> **The controller model's presentation order.** The controller is defined in two clearly separated steps rather than one. Definition 3 defines the controller purely through the failure probability $g(u) = P(F(u)=1 \mid u)$, the endorsement swing $w_\psi$, and the fallback swing $W_{fb}$, and states that no independence assumption among advisors is made at this level, and that every carrier and validity result in the paper depends only on these three quantities. Definition 4 then introduces the conditionally-i.i.d. majority-vote model as "a special parameterization" of Definition 3, and states precisely where that special case is actually used in the paper: the budget-monotonicity result and the anchored-threshold lemma. Corollary 2 then shows, by direct inspection of Theorem 1's proof, that every other result holds for arbitrarily correlated or non-identically-distributed advisors. This general-first ordering removes the apparent contradiction the review identified.

---

> > ### Author Response · Authors · 2026-07-06
> > **theoretical scope and value-evaluation assumptions**
> >
> > **Observable framing and value evaluation**
> >
> > We agree that the original "observable" wording understated the value-evaluation assumption, and we did more than soften the word. Definition 6 is retitled "Deployment-time information sets," and a new remark, Remark 1, draws the line explicitly: deployment logs -- $\mu$, $g$, the fallback identity, recorded at negligible overhead -- and value-evaluated quantities -- $W$ and $W_{fb}$, exact in a finite tabular model and otherwise bounded only via Theorem 4's construction. The remark states plainly that $\mathcal{I}_1$ and $\mathcal{I}_2$ are therefore not log-only. The same distinction now sits where a reader meets these information sets for the first time, in the Introduction and in the Abstract.
> >
> > **Scope of the $\eta=0$ regime**
> >
> > The clean $\eta=0$ regime's role was not stated prominently enough, and we fixed the framing rather than just the wording. Theorem 1 is the general carrier, valid for any $\eta \ge 0$ with both a success-side and a failure-side term; Theorem 2 is the clean failure-side characterization within it -- joint miscoordination and horizon compounding -- not the paper's only regime. We also added two results that make the general-$\eta$ case load-bearing rather than a passing remark: a range-capped certificate after Theorem 1 (Proposition 3),
> > $$
> > L \le R_{max} \wedge \left(S_\eta + C_2\right),
> > $$
> > which recovers the clean-regime certificate as $\eta \to 0$; and a tightening statement (Proposition 4) for the success side itself, showing $S_\eta$ sharpens to a realized-swing term $S_\psi$ once the executed endorsed representative is logged and its realized swing is evaluated, with a witness in Appendix A.4. In addition, Remark 2's stress test shows the resulting bound violated in every tested instance, not just a fraction, once the success-side term is dropped at $\eta = 0.2$ (Figure 10).
> >
> > **How Theorem 3 and Theorem 4 fit together**
> >
> > This relationship was previously left for the reader to work out; it is now stated in a new remark, Remark 6, placed immediately after Theorem 4's proof. The finite-sample certificate has two independent sources of error, and the two theorems are written to handle exactly one source each. Theorem 3 handles the sampling error: given any rule at all that returns a valid conservative upper bound $W_{fb}^+$ on a logged unit's swing, the empirical-Bernstein term in Theorem 3 converts $m$ independent certification episodes into a $1-\delta_B$ confidence bound on the population-level certificate. The quantity $b_0 = nHB_Q$ appearing there is only the deterministic range of the single-unit random variable being averaged, and has nothing to do with how the bound $W_{fb}^+$ was actually obtained. Theorem 4 then supplies one specific, concrete rule of that kind: it estimates each unit's swing from resettable rollouts of the environment, and inflates the estimate by a Hoeffding confidence radius, so that the resulting bound is conservative with probability at least $1-\delta_G$. Under exact tabular evaluation, no rollouts are needed at all and $\delta_G = 0$ exactly; the confidence radius that Theorem 4 adds is precisely the price paid for not having access to exact values. The two failure probabilities, $\delta_B$ from sampling and $\delta_G$ from the value bound, are then combined by a union bound into the overall $1-\delta_B-\delta_G$ guarantee stated in Theorem 3. Theorem 4's estimate is also clipped at $B_Q$, which keeps the sampled variable within the deterministic range Theorem 3 requires, and the text notes that the residual conservatism from using rollouts instead of exact values shrinks as $O(1/\sqrt{K})$ -- exact evaluation is the $K \to \infty$ limit of a construction that is already valid at every finite $K$.

---

> ### Author Response · Authors · 2026-07-06
> **empirical evidence and updated experiments**
>
> We also revised Section 9 to clarify the empirical claims. The exact-DP tabular games are used for validity auditing because they provide ground-truth value loss. The MPE experiment is used differently: it instantiates the rollout-based certificate in a learned multi-agent setting where exact ground truth is unavailable, so we do not claim an empirical tightness audit there.
>
> Section 9.1 now adds a structured resource-conflict family, where individually reasonable resource claims can jointly exceed capacity and lose the reward. The sweep covers 315 exact-DP games from $n=2$ to $n=7$ (Table 3). The certificate chain has no violations, and the operational certificate remains close to the true loss, with per-$n$ median $C_2/L$ between $1.005$ and $1.028$.
>
> Section 9.4 now applies the Theorem 3/4 rollout construction to MPE simple-spread with a trained five-member IPPO committee, using resettable rollouts rather than a learned critic or exact values. The resulting certificate reaches a ratio of 0.494 to the trivial range bound, with the decomposition in Table 4. We also included the corresponding logs and unit tests in the supplementary material, and clarified in the broader-impact discussion that a learned critic is not certified for the guarantee without separate validation.
>
> Two of the headline numbers above differ from the original submission: the rollout-bridge ratios in Section 9.2 and the MPE result just described. Both are recomputed from the released Theorem 4 implementation and its logs rather than from the earlier, less complete version, and we'd rather say so directly than leave it to be noticed.
>
> The six Requested Changes map as follows, all addressed in our first comment: (1) citations, in the Introduction's opening and the new "What kind of guarantee this is" paragraph; (2) setup motivation, the three opening paragraphs of Section 3; (3) the notation table, Table 2; (4) finite state/action spaces, Definition 1's first sentence; (5) separating claim types, the guide before Theorem 2; (6) the general-first controller model, Definitions 3-4 and Corollary 2.

---

### Review · Reviewer_5zVN · 2026-07-17

**Summary Of Contributions:**

This paper studies the committee controller in cooperative MARL, where in each round each of the $n$ agents **sequentially** takes an agent suggested by a committee: When it's agent $i$'s turn, they know round number $t$, current state $s$, and previous agents' actions; the tuple $(t,s,i,a_{<i})$ is called a "unit." At unit $u$, with probability $1-g(u)$ -- in which case the committee "endorses" an action -- agent $i$ takes action $\psi(u)$; otherwise, they take a (possibly random) fallback action $a_{fb}(u)$. All mappings, $g,\psi,a_{fb}$, are fixed but unknown.

The main result upper bounds the expected regret w.r.t. a reference policy $\pi^{\text{ref}}$. Specifically, let $\Delta_+(u,a_i):=[Q^{\pi^{\text{ref}}}(u,a_i^{\text{ref}})-Q^{\pi^{\text{ref}}}(u,a_i)]_+$ be the negative advantage of an agent $i$'s action $a_i$. The first step applies the standard PDL -- as acknowledged in Section 2, although that specific paragraph is quite confusing -- to upper bound the regret as

$nH\mathbb{E}_ {u\sim \mu}[(1-g(u))w_{\psi}(u)+g(u)W_{fb}(u)]$,

where $\mu$ is the occupancy measure induced by the **committee controller**, $w_\psi(u):=\Delta_+(u,\psi(u))$, and $W_{fb}(u):=\mathbb{E}[\Delta_+(u,a_{fb})]$ (where expectation taken over the randomness in $a_{fb}$).

This paper then tries to control it via information that's available at runtime. In the "clean" regime, the authors assume $w_\psi(u)=0$, i.e., the $\psi(u)$ is always as good as $\pi^{\text{ref}}(u)$. Three information regimes are studied:

1. $\mathcal I_0=(\mu,g)$, that is, the occupancy measure $\mu$ and the failure probability $g$ is known a-priori. In this case, a trivial upper bound $C_0=nH \mathbb{E}_{u\sim \mu}[g(u)(H-t_u)\Delta_r]$ where $\Delta_r$ is the range of rewards. I do not see anything non-trivial from this claim.
2. $\mathcal I_1=(\mu,g,W)$, where $W(u):=\max_{a_i} \Delta_+(u,a_i)$ is the worst-case advantage at any unit $u$. In this case, another bound $C_1=nH \mathbb{E}_{u\sim \mu}[g(u) W(u)]$ is established. Again, once given the definition of $W(u)$, this is straightforward.
3. $\mathcal I_2=(\mu,g,W,W_{fb})$ where $W_{fb}$ is the fallback's advantage (defined above), natually inducing $C_2=nH \mathbb{E}_ {\mu\sim u}[g(u) W_{fb}(u)]$.

This paper adds a clarification "Why the characterization is tight" **right at the beginning of this paper, when none of these notations, setups, etc are defined** (see requested changes). However, I do not see how the comment therein answers to this question. Section 6 constructs examples to make $C_0,C_1,C_2$ tight under $\mathcal I_0,\mathcal I_1,\mathcal I_2$, respectively. But this does not by itself establish that this "ladder" is the only correct notion of certificate. For example, the paper should clarify whether knowing both $\mu$ and $g$ is information-theoretically necessary, and whether knowing $W_{fb}$ -- rather than some weaker surrogate statistic -- is also necessary. The paper should also establish a **real** run-time certificate, as discussed in the next concern.

The rest of this paper argues that these $C_0,C_1,C_2$'s can be estimated from samples (Theorem 3). Numerical illustrations on the MPE benchmark is established.

**Audience:**

Yes

**Audience Explanation:**

Given the usage of committee controllers in MARL, developing a test-time certificate -- should it be really "test-time" -- would be of interest to some of the community members.

**Claims And Evidence:**

No

**Claims Explanation:**

The theorems and propositions provided in this paper are accompanied by proof (sketches). However, I do have major concerns regarding this paper's framing of their results -- which are much more bold than what are actually proved -- as summarized below:

1. This paper claims to provide a test-time guarantee. However, even the smallest information set $\mathcal I_0$ requires the knowledge of $\mu$ and $g$. The paper never provides a plug-in estimator or uniform convergence result; instead, Theorem 3 bypasses this by direct on-policy sampling. Moreover, I do not see why the $W$ and $W_{fb}$ can be known in advance, and this paper lacks justifications (Theorem 4 does mention it's possible to estimate them via resettable rollouts, but the Q-function definition relies on an exact DP or a transition simulator). This issue is not without loss of generality: Should these two functions be unknown, the only upper bound $C_0$ becomes immediately vacous.
2. This paper also fails to resolve the occupancy-measure mismatch issue between $\mu$ and $\pi^{\text{ref}}$ (which is common in offline RL or policy optimization for online RL). Namely, the expectation is w.r.t. $u\sim \mu$ -- following from the controller policy (and as discussed above, I don't believe it's reasonable to assume knowing it) -- but the functions therein is w.r.t. $Q^{\pi^{\text{ref}}}$. Therefore, if only reference-policy or historical offline data are available, the proposed "run-time" certificate does not apply without additional coverage or density-ratio assumptions.
3. See the comment regarding "Why the characterization is tight" in the previous question.
4. And finally, this paper assumes agents take sequential actions, whereas in MARL agents do take concurrent actions. It is true that one can define conditional probabilities WLOG. However, this makes it impossible to estimate $W_{fb}$ and assume it to be known: Should the $a_{fb}(u)$ be a conditional distribution, I do not see how $W_{fb}$ can be recovered from observational data, and this would (again) make $\mathcal I_2$ vacous.

**Requested Changes:**

This paper requires at least a very major revision -- if for any reason it is not direct rejected -- before it can be meaningfully reviewed. First of all, as a basic requirement of any submitted paper, i) please make sure every jargon or notation is defined before it is mentioned, and ii) please make sure the Introduction is accessible to all readers in the TMLR community.

As an example of i), nothing about "information" is mentioned before -- and even after -- the definition of $\mathcal I_0,\mathcal I_1,\mathcal I_2$. Immediately after that, the explanation of $\mu$, $g$, $W$, etc. is unspecified to a reader. I do not believe any reader who is reading sequentially can process Eq. (1) without jumping back and forth.

Regarding ii), starting from the very first paragraph of Introduction, the paper lacks sufficient justification in order for a TMLR community member -- who is not working on this paper's topic exactly -- to understand the scope, contribution, and significance. For instance, what is "test-time-compute"? What is "team value"? What is a "deployment-time certificate"? The very first paragraph even does not mention MARL, not to mention an explanation.

What is more astonishing is the paragraphs after the first paragraph of Introduction (which is by itself, as discussed above, not well-written at all). I do not quite understand why the authors put "Why this is a MARL problem.", "Why this is a sequential RL problem.", "What kind of guarantee this is.", "Information controls tightness." (and there is an enormous amount of undefined notations here; in fact, it is a common practice for this paper to use notations and jargons before they are ever defined), "Why the characterization is tight.", and "General carrier and the clean failure-side regime." Is anyone asking these questions one by one? I would suggest the authors to read academic papers and learn how an Introduction is usually written.

Aside from these clarity issues (including but not limited to the Introduction -- spanning the full draft in fact), please try to resolve the 4 issues raised above (these are by no means an extensive list; they are only the most critical issues coming up my mind after trying to decipher this paper).

---

### Review · Reviewer_wH32 · 2026-07-20

**Summary Of Contributions:**

This paper proposes a deployment-information-based certification framework for team value loss in test-time committee controllers within cooperative MARL. The authors first demonstrate that traditional per-agent marginal certification and reference-prefix telescoping bounds are invalid in cooperative settings. They then establish a range-aware information hierarchy ($C_0 \ge C_1 \ge C_2$), proving that as information becomes richer, endorsement-failure probabilities, and fallback action logs, the certificates can be progressively tightened to achieve profile-relative optimality. Furthermore, this work translates the theory’s dependence-agnostic certificate into an operational finite-sample certificate. By employing conservative value-bound constructions and variance-adaptive concentration inequalities, it achieves strictly valid and informative safety guarantees without requiring exact dynamic programming ground truth.

**Audience:**

Yes

**Audience Explanation:**

The motivation behind this work is intriguing, and it demonstrates strong novelty. Many multi-agent studies have largely overlooked the discussion of test-time compute.

**Broader Impact Concerns:**

No.

**Claims And Evidence:**

Yes

**Claims Explanation:**

The authors provide relevant definitions and theoretical proofs in their work.

**Requested Changes:**

# Strengths
1. The motivation behind this work is intriguing, and it demonstrates strong novelty. Many multi-agent studies have largely overlooked the discussion of test-time compute.
2. The work is very solid, providing rigorous theoretical guarantees.

# Weaknesses
1. I find the writing quality, particularly in the Methodology section, to be a significant issue. I suggest that the authors provide an intuitive outline or proof sketch at the beginning of this section. This would greatly enhance the readability and accessibility of the paper.
2. I have concerns regarding the scalability and effectiveness of this approach in high-dimensional state-action spaces. For instance, Theorem 4 (Conservative rollout value bound) requires $K$ rollouts for value estimation. However, in many real-world scenarios, the combinatorial space of states $s$ and action prefixes is typically vast, which could make this requirement computationally prohibitive or statistically inefficient.
3. The MPE environment used for evaluation is relatively simple. Have the authors considered evaluating their method on more complex and challenging benchmarks, such as SMAC or SMACv2, to better demonstrate its applicability?

# Summary
I find this work to be generally solid and promising. I look forward to the authors' responses to the comments raised above. Addressing these points would significantly enhance the readability of the paper and help clarify the scope and practical capabilities of the proposed algorithm.